# The genetic architecture of floral traits in the woody plant *Prunus mume*

Qixiang Zhang[1,2], He Zhang [3,4], Lidan Sun[1], Guangyi Fan[3,4,5], Meixia Ye[6], Libo Jiang[6], Xin Liu [3,4], Kaifeng Ma[1], Chengcheng Shi[3], Fei Bao[1], Rui Guan[3], Yu Han[1], Yuanyuan Fu[3], Huitang Pan[1], Zhaozhe Chen[3], Liangwei Li[3], Jia Wang[1], Meiqi Lv[3], Tangchun Zheng[1], Cunquan Yuan[1], Yuzhen Zhou[1], Simon Ming-Yuen Lee[5], Xiaolan Yan[7], Xun Xu[3,4], Rongling Wu[8], Wenbin Chen[3,4] & Tangren Cheng[1]

Mei (*Prunus mume*) is an ornamental woody plant that has been domesticated in East Asia for thousands of years. High diversity in floral traits, along with its recent genome sequence, makes mei an ideal model system for studying the evolution of woody plants. Here, we investigate the genetic architecture of floral traits in mei and its domestication history by sampling and resequencing a total of 351 samples including 348 mei accessions and three other *Prunus* species at an average sequencing depth of 19.3×. Highly-admixed population structure and introgression from *Prunus* species are identified in mei accessions. Through a genome-wide association study (GWAS), we identify significant quantitative traits locus (QTLs) and genomic regions where several genes, such as *MYB108*, are positively associated with petal color, stigma color, calyx color, and bud color. Results from this study shed light on the genetic basis of domestication in flowering plants, particularly woody plants.

[1] Beijing Key Laboratory of Ornamental Plants Germplasm Innovation & Molecular Breeding, National Engineering Research Center for Floriculture, Beijing Laboratory of Urban and Rural Ecological Environment, School of Landscape Architecture, Beijing Forestry University, Beijing 100083, China. [2] Beijing Advanced Innovation Center for Tree Breeding by Molecular Design, Beijing Forestry University, Beijing 100083, China. [3] BGI-Qingdao, BGI-Shenzhen, Qingdao 266555, China. [4] BGI-Shenzhen, Shenzhen 518083, China. [5] State Key Laboratory of Quality Research in Chinese Medicine, Institute of Chinese Medical Sciences, University of Macau, Macau 999078, China. [6] Center for Computational Biology & Key Laboratory of Genetics and Breeding in Forest Trees and Ornamental Plants of Ministry of Education, College of Biological Sciences and Biotechnology, Beijing Forestry University, Beijing 100083, China. [7] Mei Research Center of China, Wuhan 430074, China. [8] Center for Statistical Genetics, Departments of Public Health Sciences and Statistics, The Pennsylvania State University, Hershey, PA 17033, USA. These authors contributed equally: Qixiang Zhang, He Zhang, Lidan Sun, Guangyi Fan. Correspondence and requests for materials should be addressed to Q.Z. (email: zqx@bjfu.edu.cn) or to R.W. (email: rwu@phs.psu.edu) or to W.C. (email: chenwenbin@genomics.cn) or to T.C. (email: chengtangren@163.com)

Prunus mume has been domesticated for thousands of years in China because of its favorable ornamental features[1], and its cultivation has further expanded to the entire East Asia. The flowers of mei are featured for colorful corollas, varying flower types, a pleasant fragrance, and tolerance of temperatures as low as $-19\,°C$[2]. Being a long-lived woody plant, many mei trees that are hundreds or even thousands of years old in several locations in China, providing a unique set of material to study the genetic processes underlying domestication. Distant hybridization has been extensively conducted between mei and other Prunus species to improve its agronomic traits and environmental adaption, and to understand the genetic diversity of important ornamental traits[3–5]. Various types of molecular markers have been developed for mei that provide powerful tools for studying the pattern of genetic diversity within and between populations and for constructing genetic linkage maps aimed to identify QTLs controlling quantitatively inherited traits[6–9]. More recently, genome sequencing of mei has made it an ideal model system for the genetic research of woody plants[10].

Flowers have long been a focus of interest in studying this species due to their ornamental value. Mei exhibit an astonishing variety of petal colors, shapes, sizes, petal numbers, and floral bud aperture (whether there is an opening at the top of the otherwise closed floral bud). Further, other ornamental traits, such as wood color and branching habit, have also received considerable attention. Several studies have investigated the evolution of regulatory networks for transitions in floral organ identity[11,12], floral symmetry[13], and flowering time[14]. Flower pigmentation has also been studied from an evolutionary genetic perspective[14–16]. The genetic and molecular bases for the evolution of petal color have been characterized[14,15]. However, a systematic study to chart the genetic architecture of these traits in a large population using a genome-wide association (GWA) method has not yet been reported.

Here, we report the identification of significant QTLs that control floral traits in mei in a GWAS including 348 mei accessions. We further sequenced transcriptomes of flowers with diverse traits to validate the QTLs by biased expression of candidate genes between transcriptomes. The present study has for the first time elucidated the genetic architecture of floral size, color, and structure, in terms of the number of loci, their genomic distribution and the magnitude and pattern of their effects in a woody plant. In addition, by comparing mei with other Prunus species, we can begin to study the evolutionary diversification of the Prunus genome.

## Results

**Sequencing and variant discovery.** We collected a highly phenotypically diverse population including most of the existing P. mume cultivars, wild mei, and its close relatives for whole genome sequencing. A total of 348 mei accessions, including 333 landraces and 15 wild mei accessions were sampled and sequenced for the present study (Supplementary Data 1). Landraces could be classified into eleven cultivar groups (Pendulous, Single Flowered, Versicolor, Pink Double, Flavescens, Tortuosa, Green Calyx, Alboplena, Cinnabar Purple, Apricot Mei and Meiren) according to classification system of Chinese mei[1]. Three other close relatives of mei, Prunus sibirica, Prunus davidiana, and Prunus salicina, were also sampled and sequenced in this analysis. The geographic origins of the mostly representative mei accessions analyzed here spanned China, Japan, and France. Each sample was sequenced to ~19.27-fold using the Illumina HiSeq 2000 sequencing platform to generate a total of 16.28 billion raw paired-end reads and 13.71 billion clean reads after filtering (Supplementary Data 1). Deep sequencing data (~70.1-fold coverage) from eight mei trees from

different populations and three other Prunus species were used to establish the pan-genome of P. mume and Prunus genus. By mapping all clean reads against the P. mume reference genome[10], we identified a total of ~12.76 million raw SNPs and ~5.34 million high-quality SNPs after calibration and filtration (Supplementary Table 1). Applying the same variation calling method to the sequencing data from the same individual of which the genome had been assembled as ref. [10], we identified 0.28% sites to be homozygous as a different genotype which might be false positive, indicating high accuracy of the variation calling method. Allele frequency spectrum (Supplementary Fig. 1) also showed a proper segregation of the population. A total of 1,298,196 (10.17%) of raw SNPs were located within coding regions of genes, and 733,292 (5.74%) of them were non-synonymous (Supplementary Table 1). The ratio of non-synonymous to synonymous substitutions in this mei collection was 1.30, very similar to that of a collection of peach accessions (1.31)[17]. We also detected an average of 7313 deletions, 1117 insertions, and 623 structural variants (SVs) in this population (Supplementary Data 2).

**Population evolution of P. mume.** We explored the phylogenetic relationships of these 351 accessions using all high-quality SNPs identified, with three other species of Prunus genus as outgroup. The 348 mei accessions could be roughly divided into 16 subgroups within the phylogeny (Fig. 1a). We calculated bootstrap values for each node with 91.1% nodes (318/349) having a bootstrap value over 90 and all the 16 subgroups were of high confidence (Supplementary Fig. 2). We mapped ten representative ornamental traits of P. mume on phylogenetic tree and found that these traits were also quite diverse within the subgroups (Fig. 1). We carried out the population structure analysis to estimate individual ancestry and admixture proportions using FastStructure (v1.0)[18]. The population structure (Supplementary Fig. 3a) was highly-admixed and revealed eight sub-populations (when $K = 8$, the cross-validation error was minimum) (Supplementary Data 3). This was highly consistent with the phylogenetic tree (Fig. 1a) and the principle component analysis (PCA) (Supplementary Fig. 3b). The structure result was also applied in the later GWAS analysis as fixed covariate in the regression model to eliminate effects of population structure.

Linkage disequilibrium (LD) values (correlation coefficient, $r^2$) were also calculated among wild and different cultivar classes (Supplementary Fig. 3c). Higher LD was found in most cultivar sub-populations (5 out 7 cultivar classes) comparing to wild population. However, we found two classes, Pink Double and Single Flowered, to have lower LD than other cultivar sub-populations and the wild population, which should be caused by the massive introgressions from the other species to these two sub-populations (as indicated in the next session). Besides, Genetic diversity ($\pi$) of wild and cultivated mei were estimated to be $2.82 \times 10^{-3}$ and $2.01 \times 10^{-3}$, respectively, relatively low compared with crops (Supplementary Table 2). We also estimated LD decay for four representative traits (Supplementary Fig. 3d). LD levels for most subgroups were lower than that for wild group as well. For subgroups with opposite phenotypes for bud color and pistil character, LD patterns were similar. However, LD levels were higher for subgroups with red wood and green stigmas than for their corresponding opposite subgroups.

**Introgression from apricot and plum.** There were three major branches of mei cultivars (True Mume Branch originated from wild mei, Apricot Mei Branch hybrids between P. mume and P. armeniaca, and Meiren Branch hybrids between P. cerasifera cv. Pissardii and P. mume) according to previous study[1], and these

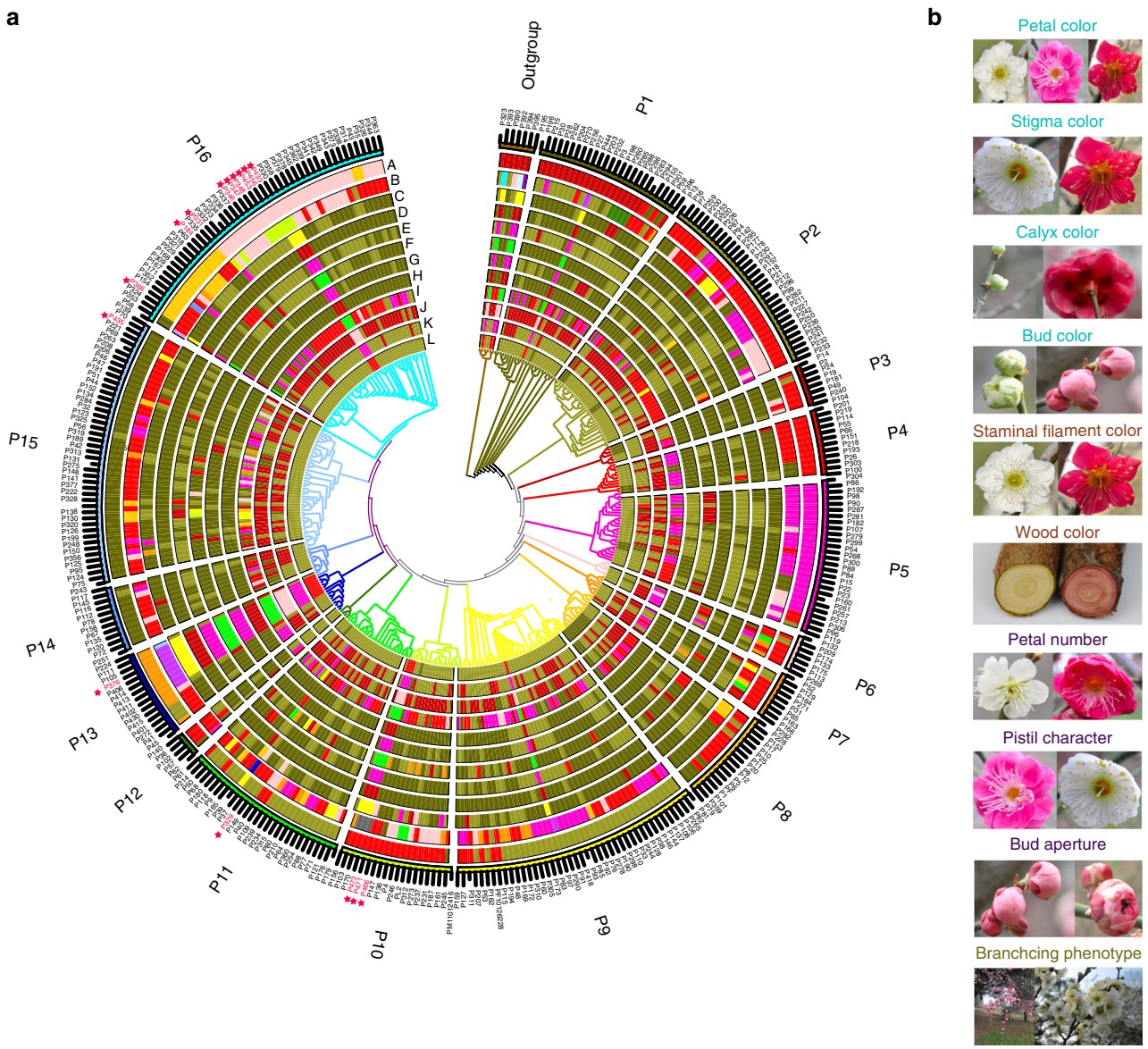

**Fig. 1** Phylogenetic tree and ten representative traits of 348 mei accessions. **a** The inner phylogenetic tree contains 16 subtrees and one outgroup. Different colors represent different subtrees. The clade color corresponds to the color of the outer circle with sample ID. The intermediate circles from the outer circle to the inner circle (A–L) represent population structure, cultivar group, and the traits petal color, stigma color, calyx color, bud color, staminal filament color, wood color, petal number, pistil character, bud aperture, and branching phenotype. The color in each circle represents the phenotype of the trait. **b** Images of several representative phenotypes of these 10 traits (photographed by T.R.C.)

three classes can be distinguished on the phylogenetic tree (Fig. 1a). Fourteen of all the 16 Apricot Mei cultivars were clustered into the outgroup or subgroup P1, which was consistent with the fact that these Apricot Mei cultivars were originated from natural hybridization between mei and apricot[19]. In the meantime, wild mei accessions appear in several lineages, while many of the current cultivars are genetically closer to apricot or Apricot Mei hybrids, which was also consistent with the artificial hybridization events between cultivars and apricot/Apricot Mei[3–5]. Taking together, there should be extensive introgression events in mei cultivars from *Prunus* species.

Further to analyze the introgression events which were considered to be essential for mei cultivation[19], we carried out the three-population $F_3$ test[20] to assess the extent of introgression (Supplementary Fig. 4, Supplementary Data 4). We first analyzed the introgression in the two hybrid cultivar groups (Apricot Mei and Meiren) and observed significant introgression from apricot

and plum, respectively (Supplementary Fig. 4), proving the reliability of the introgression analysis. Thus, then we analyzed the introgression in the nine True Mume cultivar groups (Pendulous, Single Flowered, *Versicolor*, Pink Double, *Flavescens*, *Tortuosa*, Green Calyx, *Albo-plena*, and Cinnabar Purple). Pink Double and Single Flowered (two varieties of cultivated mei) cultivars showed significant introgression (Z-score < −1.96) from the three *Prunus* species (Fig. 2), while Pendulous and Cinnabar Purple cultivars showed weak inter-species introgression signatures. This reflected the extensive inter-species introgression between mei and *Prunus* species, which made population structure and dissection of domestication history to be quite complicated.

**The pan-genome of *P. mume* and *Prunus*.** The core- and pan-genome shared by all sequenced accessions is a powerful tool for

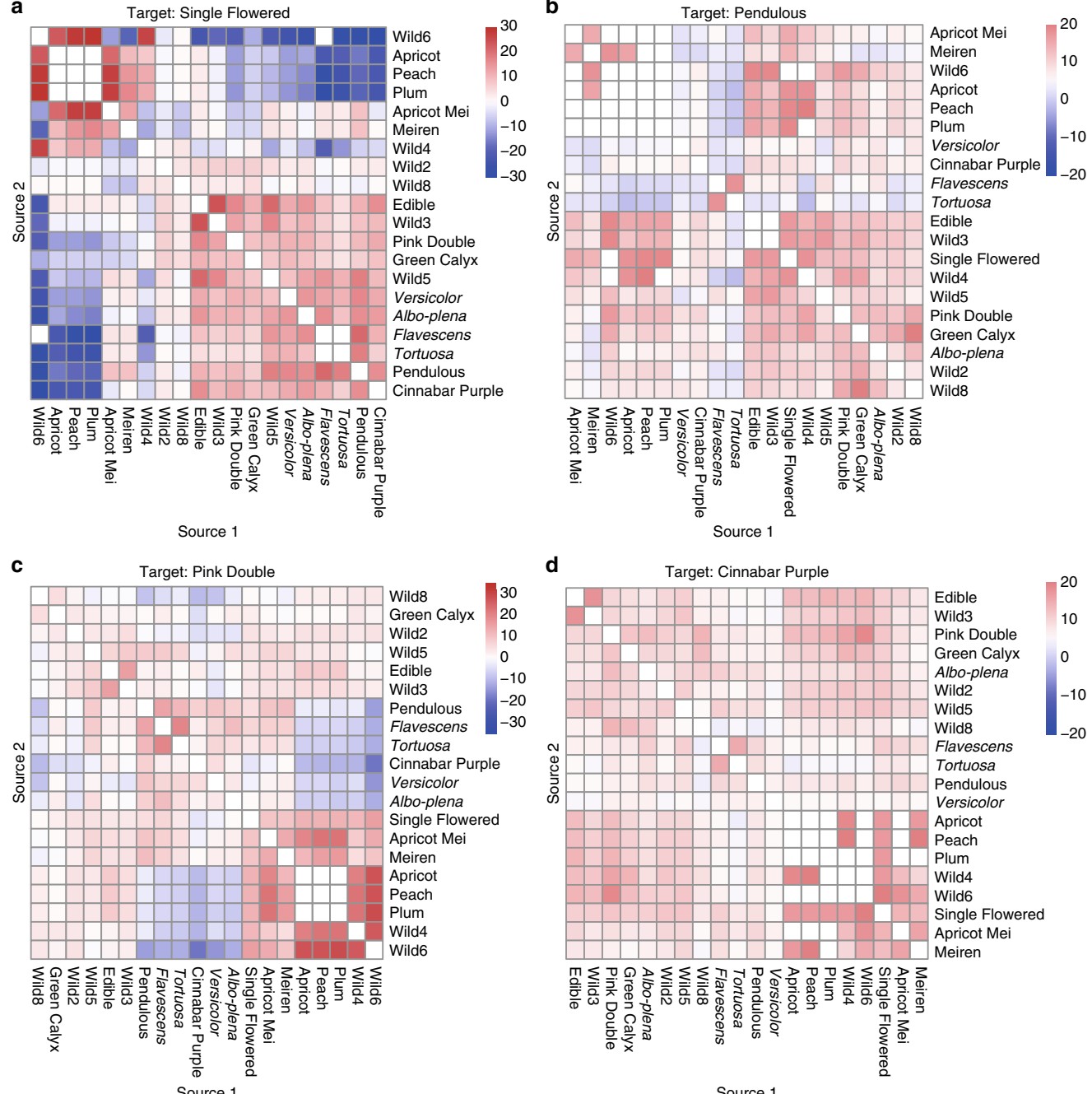

**Fig. 2** Inter- and intro-species introgression in mei population. Heat maps for three-population F3 test statistics of four cultivar groups (**a–d** represent Single Flowered, Pendulous, Pink Double, and Cinnabar Purple respectively) of mei. Introgression is significant if the Z-score is significantly negative (Z-score < −1.96, blue). Heat maps were generated using R/pheatmap package

investigating genetic diversity within populations and genomic variants arising during domestication[21]. We sequenced and assembled genomes of nine representative mei trees including those of individuals representing eight cultivars from sub-populations P2, P5, P6, P8, P9, P11, and P15 and one wild accession from subpopulation P15 (Table 1), and also those of four close relatives including *P. sibirica*, *P. davidiana*, *P. salicina* and a previously reported *Prunus persica* genome[7] to establish a pan-genome for *P. mume* and *Prunus*. To assess assembly method and evaluate assembly quality of each genome, we re-assembled the reference genome using the same approach and mapped the sequencing reads against the assembled genomes of each species. The consistent ratio (~98.13%) and sequencing depth

(Supplementary Table 3) indicated good quality of the method. Assembly of the genome from each sequenced accession resulted in an average contig N50 of ~15.5 kb and scaffold N50 of ~22.6 kb, respectively (Table 1, Supplementary Table 4). An average of 25,839 genes were annotated per genome, which accounted for 82.32% of the mei reference genome (31,390 genes in total, Table 1 and Supplementary Table 5). Assembly-based methods identified 1.30–1.47 million SNPs among the *P. mume* accessions, and 2.85–3.38 million SNPs among the other sequenced *Prunus* genomes (Supplementary Table 5). A low but consistent SNP frequency might suggest a low rate of divergence during the domestication of mei.

**Table 1 Summary of assembly and annotation of core pan-genomes of *Prunus***

| Sample ID | Origin | Assembled genome size (Mb) | Scaffold N50 (Kb) | Contig N50 (bp) | Number of genes | Number of core genes |
|---|---|---|---|---|---|---|
| *P. mume* | Tongmai, Tibet | 232.82 | 24,358.62 | 32,607 | 31,390 | 22,499[a]/19,135[b] |
| S435 | Hezhang, Guizhou | 213.79 | 23.22 | 15,510 | 25,712 | |
| S159 | Nanjing, Jiangsu | 213.27 | 20.52 | 13,942 | 25,926 | |
| S176 | Wuhan, Hubei | 208.17 | 23.45 | 16,128 | 25,645 | |
| S1 | Wuhan, Hubei | 215.74 | 19.73 | 13,737 | 25,533 | |
| S200 | Suzhou, Jiangsu | 211.53 | 21.98 | 14,682 | 25,484 | |
| S209 | Japan | 208.43 | 32.05 | 19,783 | 25,575 | |
| S248 | Chongqing | 219.68 | 17.42 | 12,260 | 26,593 | |
| S89 | Wuhan, Hubei | 225.95 | 27.97 | 19,732 | 26,754 | |
| S93 | Huizhou, Anhui | 210.87 | 21.26 | 14,109 | 25,521 | |
| *P. salicina* | NA | 210.25 | 14.67 | 10,307 | 24,294 | |
| *P. davidiana* | NA | 237.17 | 21.98 | 15,674 | 26,726 | |
| *P. sibirica* | NA | 217.96 | 26.50 | 19,703 | 26,303 | |
| *P. persica* | NA | 227.25 | 26,807.72 | 214,242 | 27,792 | |

[a]Core gene number identified among *P. mume* individuals
[b]Core gene number identified among *Prunus* individuals

We found that 71.68% and 60.96% of the *P. mume* reference genes were shared by all nine *P. mume* and 13 individual *Prunus* accessions (Supplementary Tables 6 and 7). We found that 3364 genes did not appear in the *Prunus* core gene set and were thus mei-specific genes (Table 1). Among the mei-specific genes, those related to flavonoid, phenylpropanoid, stilbenoid, diarylheptanoid and gingerol biosynthesis, and phenylalanine metabolism were relatively enriched (Supplementary Data 5). These functional pathways likely influence the development of important aspects of the ornamental traits of mei. For instance, flavonoids are important plant pigments that influence flower coloration, and phenylpropanoids serve as essential components of floral pigments and scent compounds in addition to their roles in wood formation. Our findings were further confirmed by the subsequent GWAS. It is noteworthy that three out of six *DAM* (dormancy-associated MADS-box transcription factors family) genes reported in *P. mume*[10] were not present in either the mei or *Prunus* core gene sets, possibly due to the divergence of blooming time between mei and *Prunus* accessions.

Novel sequences in each genome were identified by whole-genome alignment with the reference genome and the core pan-genome assembly. Detected presence-absence variations (PAVs) varied in length from 0.19 to 0.55 Mb in *P. mume* genomes and from 8.94 to 25.85 Mb in other *Prunus* genomes (Supplementary Tables 8 and 9). To eliminate low-confident PAVs and illuminate their population patterns, we mapped reads from 351 resequencing samples to all eight *P. mume* genomes' PAVs. We found about 6.25% of PAVs resulted from unassembled sequences and thus excluded them from following analysis. We further performed hierarchical clustering of samples based on the distribution of coverage and identity of high-confidence and population-specific PAV sequences (Supplementary Table 10). Samples could be clustered into ~16 groups, in high concordance with the number of previously identified subpopulations (Fig. 3a). We then discovered several population-specific PAVs that showed particular patterns within geographical subpopulations, and decreased coverage that might reflect domestication. Subpopulation P11 includes the wild mei accession S329 from Tibet, which is regarded as the progenitor of current mei accessions, and S179, which were used to construct the *Prunus* core pan-genome. Two PAVs identified in the S179 genome were highly specific to subpopulation P11 and showed different patterns of change in coverage during domestication (Supplementary Table 11). For example, coverage of PAVs was high in accessions from Tibet, Wuhan, and other populations from south China, but coverage was lower in populations from north China, including Beijing and Qingdao, and those from Japan (Fig. 3b). PAVs could thus be used for identification of mei accessions.

We used core genes identified in 13 individual *Prunus* accessions and three sequenced relatives from the Rosaceae to construct a phylogeny and reconstruct the evolutionary history of *Prunus* genus. Our results suggested that *P. sibirica* might be more closely related to mei than is any other *Prunus* species in the present study, which was consistent with a previous report[22]. We estimated the divergence times between *P. mume* and other *Prunus* species as ~3.8 MYA and between wild and cultivated mei as ~2.2 MYA, which far predated the estimated domestication of cultivated mei (Fig. 3c). Therefore, divergent selection may have contributed to the differentiation of the two subspecies long before the domestication of mei.

**GWAS analysis of ten ornamental traits.** Our study population was composed of different landraces of a woody plant, whose selection had a different process from what is true in crops, as mentioned above. Thus, we developed a method based on logistic regression to perform GWAS of 24 mei floral traits (Supplementary Data 3), considering population structure calculated before as a fixed covariate in the model. It turned out that our method provided quite satisfactory results about GWAS, as shown by Q–Q plots of *p*-values for each trait (Supplementary Fig. 5), in which *p*-values after correction were found to be close to expected curves. We identified five significant (Bonferroni-corrected $P < 0.01$) candidate regions on four chromosomes associated with ten traits including petal color, stigma color, calyx color, bud color, staminal filament color, wood color, petal number, pistil character, bud aperture, and branching phenotype (Fig. 1b). To validate the differential expression of candidate genes significantly associated with flower-related phenotypes in mei, we performed RNA-Seq analysis by sequencing six transcriptomes of two representative cultivars, 'Wu Yu Yu' and 'Mi Dan Lv' (WYY and MDL for short, three biological replicates per sample) that have distinct phenotypes for petal, calyx color, and petal number (Supplementary Table 12). We identified a total of 3277 significant differentially expressed genes (DEGs), 159 of which were specifically expressed in one sample type and subsequently designated as specifically expressed genes (SEGs) (Supplementary Fig. 6).

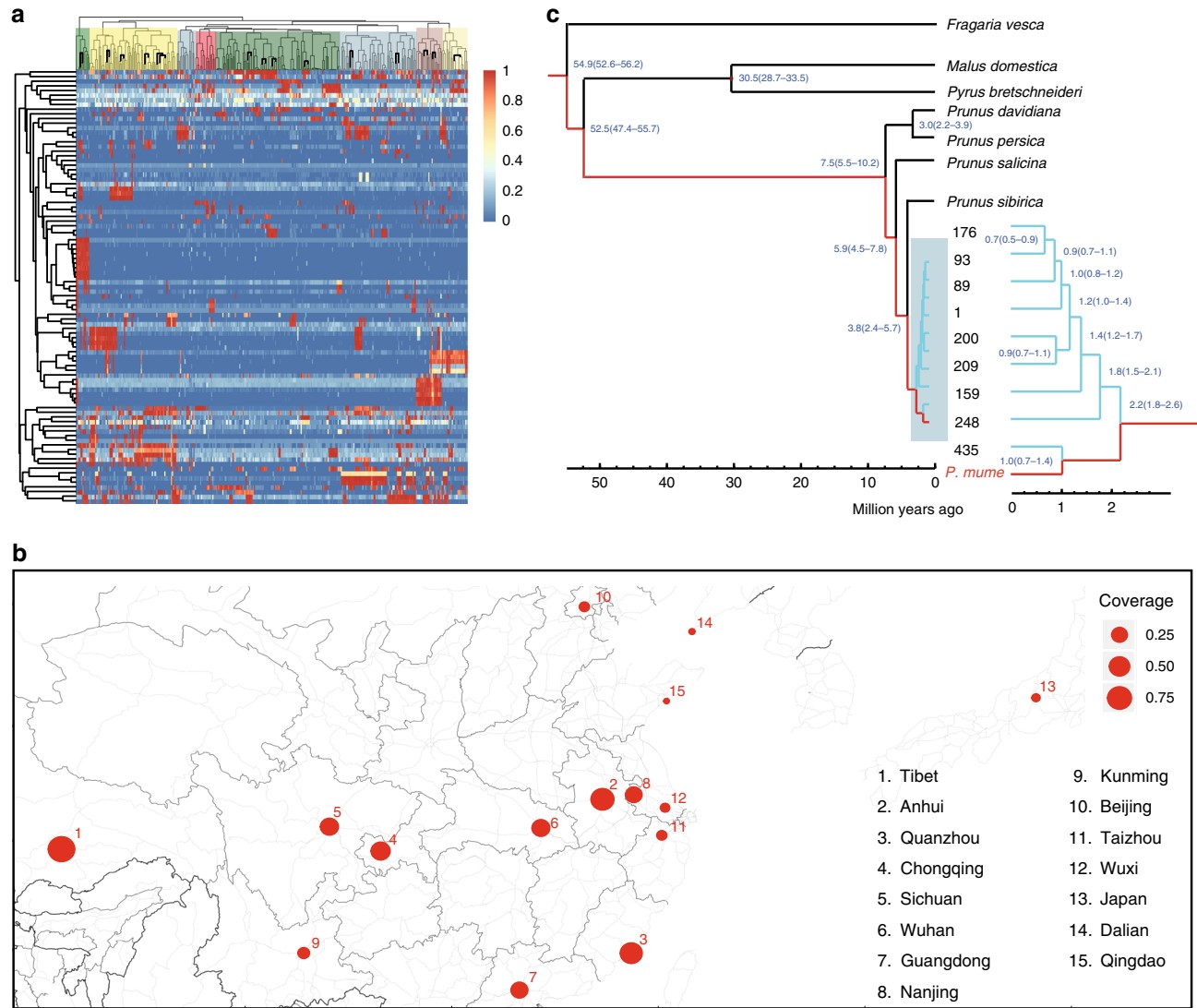

**Fig. 3** Evolution of *P. mume* and *Prunus*. **a** Clustering of 351 re-sequencing samples using coverage of PAVs in mei genomes. Coverage ratio in each box was calculated by mapping reads from each sample to each PAV. **b** A PAV identified in S176 shows divergent coverage across samples in subpopulation P11. Map image was generated from http://maps.stamen.com/#toner/12/37.7706/-122.3782, using R/ggmap package. **c** Reconstruction of the evolutionary history of *Prunus* and *P. mume* accessions

We identified a total of 76 SNPs within DEGs that were associated with petal, stigma, calyx and bud color, respectively (Supplementary Data 6-9). Interestingly, we found that these SNPs were associated with the region spanning from 229 kb to 5.57 Mb on chromosome Pa4 (Fig. 4a, Supplementary Figs. 7, 8 and 9). *MYB108* (Pm012912, Pa4:411731:413009) encodes an R2R3 MYB transcription factor and located in the same region on chromosome Pa4 as the petal color-associated SNPs (Fig. 4a). Several members of this gene family reportedly affect flower color in plants[23–25]. Moreover, *MYB108* was expressed in all three WYY samples, which have red flowers, but not in any of the MDL samples, which have white flowers (Fig. 4b). To investigate the regulatory network affecting expression of *MYB108*, we used the String database (http://string-db.org/)[26] to construct an inter-active network of the SEGs identified above (Fig. 4c). In the network model, the E2F transcription factor 1 (E2F1) has the same expression pattern as *MYB108*, which binds a sequence present in the promoter of the S-phase-regulated gene CDC6 and is a member of a multigene family with several different activities

in *Arabidopsis*[27]. This indicated that E2F1 might be involved in the regulation of *MYB108* gene expression. We also predicted three possible promoters in the upstream region of *MYB108* using TSSP[28] and found that one of them was significantly associated with a SNP (Pa4:411479, $P = 11.39$) located at an ambiguous nucleotide in this promoter (RSP00161, WAAAG, Supplementary Data 10). This mutation in the putative *MYB108* promoter might affect the regulation of *MYB108* expression. We also found that *MYB108* only exists as a single-copy gene in all reported plant genomes. *MYB108* is highly conserved within *Prunus*, but differs notably among clades (Supplementary Figs. 10 and 11). These results suggest that *MYB108* might play a critical role in the control of petal color in *Prunus*.

Our GWAS results showed that wood and staminal filament color were each significantly associated with two regions located on chromosome Pa3 (wood color, R1: 20601577-20832908 and staminal filament color, R2: 444623-3375607, Fig. 5a, Supplementary Fig. 12, Supplementary Data 11–12). R1, which spans ~231 kb and contains 48 genes, includes SNPs significantly

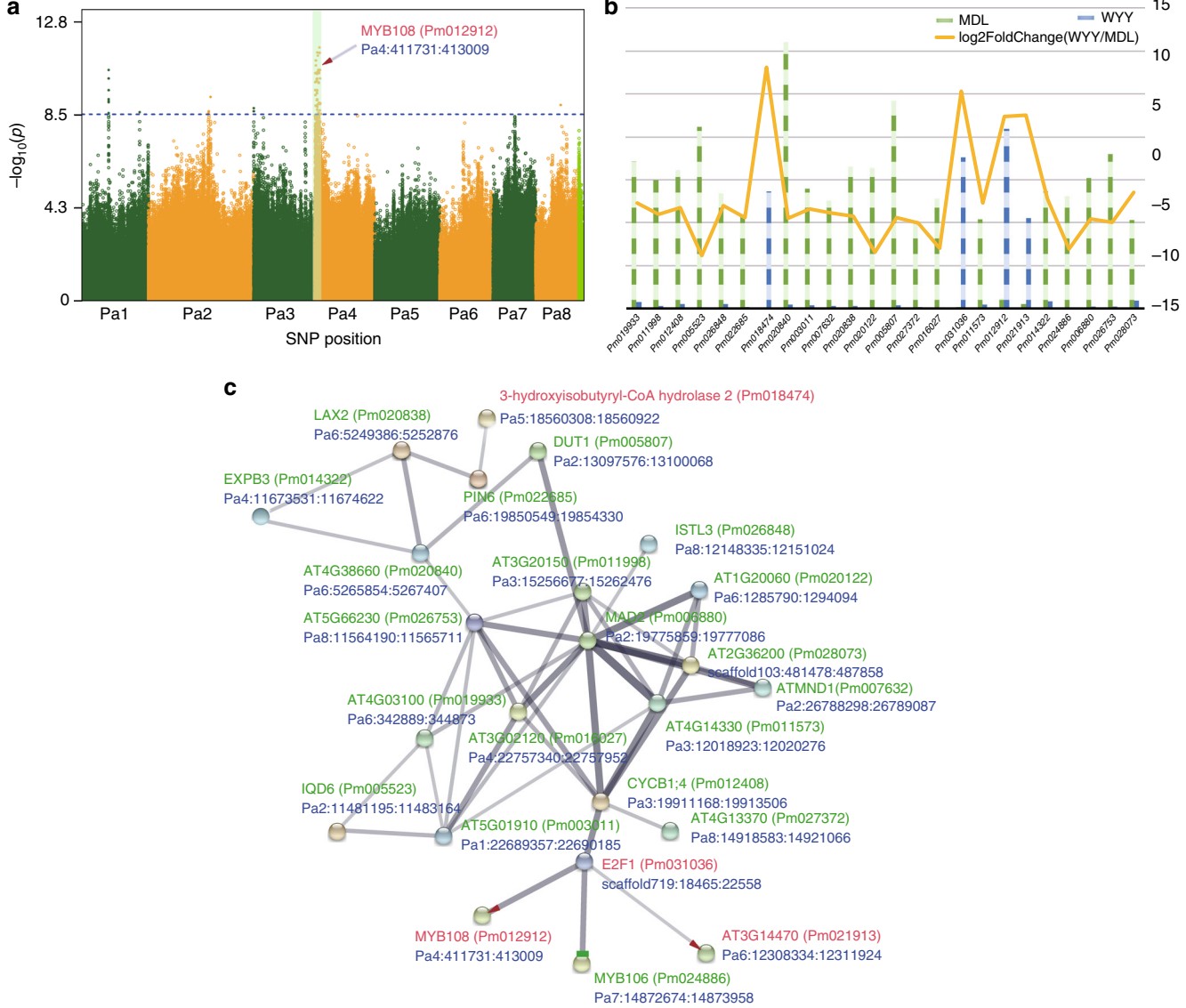

**Fig. 4** GWAS results of petal color and other three floral traits. **a** Manhattan plot of the petal color trait on the eight chromosomes of mei. **b** Comparison of the transcript expression of SEGs located in the Pa4 region significantly associated with petal color, stigma color, calyx color, and bud color. **c** The co-expression network of SEGs based on the interaction relationships of *Arabidopsis thaliana* orthologs in the String database

associated with wood color compared to those in the regions flanking R1 (Fig. 5b). We also identified a polymorphism in R1 between subpopulations with green or red wood by calculating the Fst[29] and π values for these subpopulations, which indicated that R1 was involved in a selective sweep driven by artificial selection[30] (Fig. 5c). For staminal filament color, we focused on the significant DEGs between MDL and WYY in these two regions (Fig. 5d). In R2, we found several candidate genes that could affect staminal filament color, including *HAESA* (Pm010102) (Fig. 5e), a dual-specificity receptor kinase that acts on both serine-/threonine- and tyrosine-containing substrates to control floral organ abscission[31]. Another is *SPL5* (squamosa promoter binding protein-like 5, Pm010075), a transacting factor that binds specifically to a consensus nucleotide sequence in the AP1 promoter[32]. A third is *PRXR1* (peroxidase 42, Pm009792), which is involved in the biosynthesis and degradation of lignin[32].

We identified a region spanning ~3.64 Mb located on chromosome Pa1 (4058003-7693997) that was significantly

associated with petal number, pistil character, and bud aperture of mei (Supplementary Figs. 13, 14 and 15, Supplementary Data 13–15). This region contained 41 DEGs, including the SEGs Pm000751 (LAC17, laccase 17), Pm001026 (hydroxyproline-rich glycoprotein family protein), and Pm000753 (PRS, PRESSED FLOWER) (Supplementary Data 16). LAC17 functions in lignin degradation and detoxification of lignin-derived products[32–35]. PRS is a probable transcription factor that is involved in lateral sepal axis-dependent flower development perhaps by regulating the proliferation of L1 cells in lateral regions of flower primordia[33,36,37]. These two genes could be involved in the control of petal number and pistil character of mei.

A candidate region associated with branching phenotype spanning ~1.15 Mb located on chromosome Pa7 was identified in a hybrid population in a previous study[38]. In our study, several candidate regions were also found on chromosome Pa7 (Supplementary Fig. 16). Between the two studies, a total of 13 candidate genes that might be associated with branching

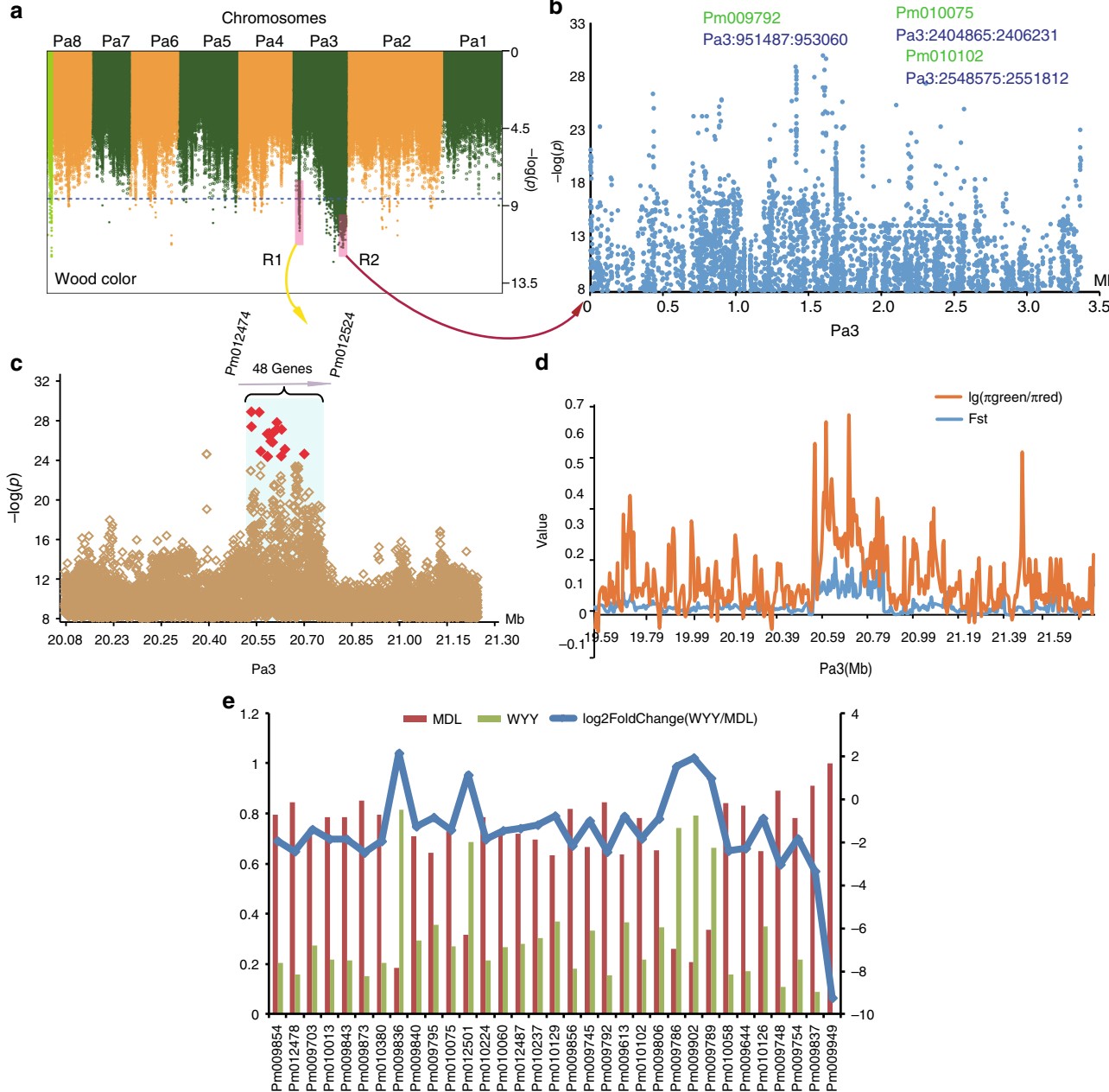

**Fig. 5** GWAS results of wood and staminal filament color traits. **a** Manhattan plot of the wood color trait on the eight chromosomes of mei. **b** The region significantly associated with staminal filament color, designated as R2. **c** The region significantly associated with wood color, designated as R1, contained 48 candidate genes. **d** Comparative analysis of SNP polymorphisms between green wood and red wood groups. Values for π and Fst were calculated to identify possible selective sweep regions. **e** Comparison of the transcript expression of SEGs located in the R1 region significantly associated with wood color and staminal filament color

phenotype, including the transcription factor bHLH157 (Pm024214) and cytochrome P450 78A9 (*CYP78A9*, Pm024229), have been identified (Supplementary Data 17). Several P450-dependent reactions have been characterized in the phenylpropanoid pathway, which controls the synthesis of lignin[39].

**Discussion**

By using a combined strategy of genome resequencing, de novo assembly and genome-wide association analysis, we address a fundamental challenge for inferring the population diversification of mei (*Prunus mume*), a woody plant, and the genetic architecture of its floral traits. We sampled and sequenced 333 mei landraces and 15 individual mei trees from a wild stand, along with three other *Prunus* species, to reveal the genetic divergence of mei under its domestication. We studied the genetic control of flower color, structure, and shape because these traits directly determine the reproductive behavior, mating system, and domestication process of higher plants[40].

The 348 mei accessions sampled were clustered into 16 distinct subgroups, suggesting mei landraces in our population might be selected from wild group and could be regarded as founder (Fig. 1). Our findings testified extensive introgression between

mei and other *Prunus* species, even in some True Mume groups, according to the most-update classification system of mei (Fig. 2). Moreover, we found two cultivar groups had lower LD level than the wild and these two groups showed strong signature of introgressions from *Prunus* species. It suggested that introgression was probably the reason why higher LD was observed in wild population, as compared to that in these cultivar groups. These evidences together lead to complicated ancestry of current mei population.

By further deep sequencing of nine representative landraces of *P. mume* and its relatives, *P. persica*, *P. salicina*, *P. sibirica* and *P. davidiana*, we established a pan-genome for *Prunus*. Lineage- and species-specific sequences were found to highly correlated with sub-population genetic architecture (Fig. 3) or particular adaptive or ornamental traits. These data will facilitate future work in evolutionary research of *Prunus* genus and genetic basis of floral traits.

The genetic control of flower-related morphologies of mei was found to be polygenic. For each of the ten floral traits studied, including petal color, wood color, petal number, flower density, branch type, and branching structure, numerous QTLs were identified (Figs. 4 and 5, Supplementary Data 18), but with a single locus explaining 1–3% of the phenotypic variance. Further gene expression studies characterized the transcripts of genes that underlie these QTLs. Many of the QTLs detected were found to reside in genomic regions near candidate genes. A QTL affecting petal color located in the same region of *MYB108* (Pm012912, Pa4:411731:413009) on chromosome Pa4. *MYB108*, encoding an R2R3 MYB transcription factor, was observed to mediate flower color in plants[23–25]. By combining transcriptome data and data from existed database, we further reconstructed a regulatory network related to MYB108, from which the E2F transcription factor 1 (E2F1) that binds a sequence present in the promoter of the S-phase-regulated gene CDC6 was found to activate the expression of *MYB108*. As a highly conserved gene within *Prunus*, *MYB108* may have played a critical role in modulating the genetic control and evolution of petal color in *Prunus*.

We identified two QTLs located on chromosome Pa3, associated with wood color and staminal filament color, respectively (Fig. 5a, Supplementary Fig. 12, Supplementary Data 11-12). A polymorphism within the first QTL was detected to determine how wood is colored. One genotype at this polymorphism forms green wood, whereas the alternative forms red wood. By calculating the Fst[29] and π values for the subpopulations that are dominated by two alternative genotypes at this polymorphism, it was inferred that the QTL for wood color may have been involved in a selective sweep driven by artificial selection[30] (Fig. 5c). The QTL for staminal filament color are associated with several candidate genes, including *HAESA* (Pm010102) (Fig. 5e), a dual-specificity receptor kinase that acts on both serine-/threonine- and tyrosine-containing substrates to control floral organ abscission[31], squamosa promoter binding protein-like 5 (*SPL5*, Pm010075), a transacting factor that binds specifically to a consensus nucleotide sequence in the AP1 promoter[32], and *PRXR1* (peroxidase 42, Pm009792), which is involved in the biosynthesis and degradation of lignin[32].

Petal number is a key trait that determine the flower structure of mei. One QTL for petal number, as well as for pistil character and bud aperture, was found in a region of chromosome Pa1 (Supplementary Figs. 13, 14 and 15, Supplementary Data 13–15), containing 41 DEGs, such as the SEGs Pm000751 (LAC17, laccase 17), Pm001026 (hydroxyproline-rich glycoprotein family protein) and Pm000753 (PRS, PRESSED FLOWER) (Supplementary Data 16). These genes display important functions in lignin degradation and detoxification of lignin-derived

products[32–35] and lateral sepal axis-dependent flower development.

The candidate regions associated with petal color (Pa3), wood color (Pa4), and petal number (Pa1) located in different regions of the mei genome, and their independence suggest that these traits have experienced different routes of evolution. Although many QTLs detected can be annotated, there are also many QTLs that are in previously uncharacterized regions or not associated with any annotated genes. The biological functions of these unknown QTLs deserve further investigation. Taken together, the identification of genetic loci associated with floral and other traits provides more insight into the genetic mechanisms that underlie the domestication of mei and provides opportunities to design strategies for genomic selection to improve the performance of ornamental species.

## Methods

**Plant materials and genome sequencing.** We collected leaves from 333 representative mei landraces, 15 wild *P. mume*, and three close relatives of *Prunus*, including *P. sibirica*, *P. davidiana*, and *P. salicina* (Supplementary Data 1). Meanwhile, information for up to 24 phenotypic categories was recorded for each sample (Supplementary Data 3). Genomic DNA was extracted from fresh or gel-dried leaves using a standard cetyl trimethyl ammonium bromide (CTAB) protocol[41]. After DNA extraction, genomic libraries were prepared following the manufacturer's standard instructions (Illumina). To construct paired-end libraries, DNA samples were fragmented by nebulization with compressed nitrogen gas and treated to create blunt ends before adding an A to each 3′-end. DNA adaptors with a single T 3′-end overhang were ligated to the above products. Libraries with short insert sizes of 500 bp were constructed for each resequencing sample, and extra libraries with long insert sizes of 2 kb were constructed for the core- and pan-genome assembly. All libraries were sequenced on the Illumina HiSeq 2000 sequencing platform, and paired-end reads from each library were obtained. Reads were firstly filtered to obtain high-quality sequences for mapping and assembly by eliminating: (i) reads containing ≥10% ambiguous bases, (ii) reads with low-quality data (Phred quality scores $Q ≤ 7$) for 65% of bases for short inserts libraries (<2 kb) or 80% of bases for long inserts (2 kb), (iii) reads containing 10-bp adaptor sequences, (iv) reads with >10 bp overlaps between two ends of short-insert reads, and (v) reads with identical sequences at both ends.

**RNA-Seq of accessions with contrasting phenotypes.** We performed RNA-seq for fresh flowers from two mei landraces (MDL vs. WYY) with three replicates for each were collected in April, 2016. Samples were immediately fixed in liquid nitrogen after collection and stored at −80°C. Total RNA was isolated using a modified CTAB protocol[42] and used for cDNA library preparation. RNA was first assessed by capillary electrophoresis on an Agilent BioAnalyzer 2100 (Agilent Technologies, Palo Alto, California, USA). Polyadenylated RNA isolated using oligo (dT)-attached beads was fragmented and reverse transcribed to cDNA. Paired-end libraries, with 500 bp in length, generated from each sample were then sequenced separately. RNA-seq were performed on the Illumina Hiseq 2000 platform. A total of 483.58 Mb raw data were produced by transcriptome sequencing of six samples and details were mentioned in Supplementary Table 12.

**Variant identification.** All reads were mapped against the mei reference genome using BWA[43]. SNP calling was then performed following best practices for the GATK[44] (v3.1) pipeline (using mainly the module UnifiedGenotyper, followed by quality filtering (VariantFiltration with parameters: QD < 2.0 || FS > 60.0 || MQ < 40.0 || HaplotypeScore > 13.0). We further filtered SNPs with more than 10% samples with missing genotypes, or deviated from Hardy-Weinberg principle in some downstream analysis. Structural variants (SV) were identified using Break-Dancer[45], which cataloged deletions, insertions, inversions, and intra-chromosomal translocations.

**Population analysis.** Genetic distance was estimated using genome-wide SNPs[46], where the distance between two individuals $i$ and $j$ was defined as:

$$D_{ij} = \frac{1}{L} \sum_{l=1}^{L} d_{ij}^{(l)}, \tag{1}$$

$L$ represented the length of the SNP region, where at position 1, $d_{ij}$ would be equal to 0 or 1 if genotypes were identical or different between two individuals, and $d_{ij}$ would be equal to 0.5 for other scenarios. A matrix of genetic distance was then used to generate a Neighbor-Joining tree using PHYLIP (v3.69)[47]. Linkage disequilibrium values for wild and cultivated mei, and populations with particular traits were calculated using Haploview[48] with filtration parameters "-minMAF 0.05 -hwcutoff 0.01." LD decay was then estimated as the relationship between the

distance between each pair of SNPs and their corresponding correlation coefficient ($r^2$) value. The distance that the LD decays to half of the highest value was then calculated. Principle Component Analysis were taken by EIGENSOFT[49] software using genome-wide SNP.

**Genome assembly and establishment of a core- and pan-genome**. Nine mei and three *Prunus* varieties were subjected to deep sequencing (~70.1×) using a single paired-end library and a mate-paired library with insert sizes of 500 bp and 2 kb, respectively, using the Illumina HiSeq 2000. All reads from each sample were processed and assembled using SOAPdenovo[50](v2.04), followed by gap closing using GapCloser (v1.12) from the SOAPdenovo package. Both homology-based and ab initio prediction of protein-coding genes was then performed using *P. persica*, *P. mume*, *Fragaria vesca*[51] and *Pyrus × bretschneideri*[52] gene sets as references, and final gene prediction results for each genome were generated using GLEAN[53]. The core-genome was established by: (1) aligning all 12 assembled genomes plus the *P. persica* reference genome to the *P. mume* reference genome, using pairwise alignment software NUCmer from the MUMmer package;[54] (2) filtering alignment results with identity <90%, map_len_min = 100 bp, query_seqs <500 bp and coverage <0.8; and finally (3) extracting and retaining core-genome sequences for nine mei and 13 *Prunus* species under different ratio.

**Specific sequences identification**. PAVs in each genome assembly were identified in a stepwise manner. First, unaligned sequences were collected after aligning all mei and *Prunus* genomes analyzed here during the establishment of the core- and pan-genome. Then these previously unaligned sequences were realigned to the reference genome using BLAT[55] and unaligned sequences with identity <95% and length >100 bp were extracted. Finally, sequences generated from each assembly above were aligned to any other assemblies to identify any sequences specific to individuals (identity <90%, length >100 bp). For validation of the PAVs and identification of population-specific PAVs, we mapped all paired-end reads from the 351 resequenced samples and calculated coverage for each PAV. From this set, PAVs covered by more than 90% of the samples with coverage and identity over 90% were excluded. To investigate population pattern of PAVs, a total of 93 PAVs were selected and their distribution among all samples was displayed on heatmap using R/pheatmap[56]. PAVs specific to the P11 subpopulation were then identified and average coverage was calculated in the genomic vicinity of each PAV. The ggmap[57] package in R was used to visualize the distribution of PAVs across these locations.

**Phylogenetic analysis**. The phylogenetic tree for mei and other chosen plant genomes was constructed using single-copy orthologous genes identified using the clustering program OrthoMCL[12]. The generation-time hypothesis, which suggests that shorter generation times could accelerate a species' molecular clock[58], might explain differences in the divergence rates in molecular clocks. Firstly, we used MUSCLE[59] to align the predicted proteins of single-copy genes. Then the protein sequences were reverse-transcribed into coding DNA sequences (CDS) based on the alignment results. Fourfold degenerate sites in each alignment were identified and concatenated into a single supergene for each species. A phylogenetic tree was then constructed using PhyML[60] or MrBayes[61]. The fourfold degenerate sites were used to estimate the neutral substitution rate per year and divergence times among species.

The models "Correlated molecular clock" and "JC69" were chosen to calculate species divergence times using the MCMCTREE program from the PAML[62] package. The Markov Chain Monte Carlo (MCMC) process in MCMCTREE program was run 800,000 times, with a burn-in of 80,000 iterations. Four independent runs were performed to check convergence.

**Genome-wide association study**. In this GWAS study[63,64], we describe the measured value $y_i$ of a trait for individual $i$ using a regression model

$$y_i = \mu + X_i\beta + \xi_i a + \zeta_i d + \varepsilon_i \qquad (2)$$

where $\mu$ is the overall mean, $X_i$ refers to the subpopulation of individual $i$, $\beta$ is the effect for subpopulation, $a$ and $d$ are the additive and dominant effect of SNPs, respectively, $\xi_i$ and $\zeta_i$ are the indicator vectors of the additive and dominant effects of SNPs for individual $i$, and $\varepsilon_i$ is the residual error. The $J$-th elements of $\xi_i$ and $\zeta_i$ are defined as

$$\xi_{ij} = \begin{cases} 1, & \text{if the genotype of SNP } j \text{ is AA} \\ 0, & \text{if the genotype of SNP } j \text{ is Aa} \\ -1, & \text{if the genotype of SNP } j \text{ is aa} \end{cases} \qquad (3)$$

$$\zeta_{ij} = \begin{cases} 1, & \text{if the genotype of SNP } j \text{ is Aa} \\ 0, & \text{if the genotype of SNP } j \text{ is AA or aa} \end{cases} \qquad (4)$$

The regression coefficients can be fitted using maximum likelihood method. The hypothesis about a marker affecting trait can be formulated as

$$H_0 : a = 0, d = 0$$
$$H_1 : \text{at least one of the above does not hold,}$$

where $H_0$ corresponds to the reduced model and $H_1$ corresponds to the full model. The test statistics for testing the hypothesis is calculated as the log-likelihood ratio (LR) of the full over reduced model. LR can be viewed as being asymptotically $\chi^2$ distribution with different df. Level of significance is adjusted by Bonferroni method.

However, in GWASs, a number of traits are either discrete or continuous. The discrete traits are categorized into binary, multinomial, and ordinal. Traits such as white or red flowers or calyx color due to pigment phenotypes are classic unordered categorical variables. This study used three different methods to detect relationships between genotypes and phenotypes based logistic regression model for discrete traits. (1) General linear model (GLM): GLM was used to test the associations based binary logistic regression link function for the binary variable. (2) Multinomial logistic model (MLM): MLM was used that the dependent variable is nominal with more than two levels and no intrinsic ordering. (3) Ordinal logistic regression (OLR): OLR was used to test the associations between markers and multiple ordered levels. For the continuous traits, linear regression model was used to test association.

Besides, we used two models to analyze the data, considering effects of population structure or kinship. The first is the Q model that adjusts for the population structure of mei varieties, and the second is the Q + K model that correct for population structure and the kinship, i.e., the most probable identity by state of allele between varieties. The degree of kinship was calculated through correlation analysis using marker data. The model that better fits the data was assessed and chosen according to the Q−Q plot, in conjunction with the inflation factor estimated from the median of the test statistics for all the markers. Optimal model for each trait and Q−Q plot with results generated from both models were presented in Supplementary files (Supplementary Table 13, Supplementary Fig. 17).

**Functional enrichment analysis**. We performed functional enrichment analysis of genes associated to floral traits, with the Gene Ontology (GO), and Kyoto Encyclopedia of Genes and Genomes (KEGG) databases. By comparing with the background of all genes in mei genome using hyper-geometric distribution, enrichment analysis provides all terms (GO term and KEGG pathway ID) that are significantly enriched in the target genes. *P*-value was thus defined as

$$P = 1 - \sum_{i=0}^{m-1} \frac{\binom{M}{i}\binom{N-M}{n-i}}{\binom{N}{n}} \qquad (5)$$

where $N$ is the count of all genes with function information in mei genome; $n$ is the count of genes associated to specific floral traits in $N$; $M$ is the count of all genes annotated to certain functional terms or pathway; and $m$ represents the count of genes associated to specific floral traits in $M$. The calculated p-value experienced Bonferroni Correction, and functional terms or pathways with corrected p-value ≤ 0.05 were defined as significantly enriched functional terms.

**Code availability**. Computer code used to perform population genomics and GWAS is available from the corresponding authors upon request.

**Data availability**. The sequence data of mei genome resequencing involved in this study have been deposited in NCBI with the accession number SRP093801 (Bio-Project: PRJNA352648). All other relevant data supporting the findings of the study are available in this article and its Supplementary files, or from the corresponding authors upon request.

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

## Acknowledgements

The research was funded by the National Natural Science Foundation of China (Grant No. 31471906), the Fundamental Research Funds for the Central Universities (No. 2016ZCQ02), the Special Fund for Beijing Common Construction Project, and the

National High Technology Research and Development Program of China (2013AA102607).

## Author contributions

Q.X.Z., T.R.C., W.B.C., X.X., and R.L.W. conceived the study. L.D.S., X.L.Y., Y.H., and Y.Z.Z. collected all materials and phenotypic data. G.Y.F., H.Z., C.C.S., R.G., X.L., M.X.Y., L.D.S., Y.Y.F., L.W.L., Z.Z.C., L.B.J., and M.Q.L. contributed to SNP calling, population analysis and GWAS. L.D.S., K.F.M., H.T.P., J.W., C.Q.Y., T.C.Z., and F.B. performed experimental work. H.Z., S.M.Y.L., L.D.S., R.L.W., X.L., and G.Y.F. wrote the manuscript with input from other authors. All authors approved the manuscript before submission.

## Additional information

**Competing interests:** The authors declare no competing interests.

