## [Peer Review File · Nature Communications]

Reviewers' comments:

Reviewer #1 (Remarks to the Author):

Comments for the manuscript entitled "The genetic architecture of floral traits in the woody plant *Prunus mume*"

Mei is an ornamental woody plant in East Asia. The authors selected and sequenced 348 accessions to conduct the comprehensive diversity study of mei. In addition, a total of nine accessions were used to construct the pan-genome via deep sequencing. GWAS was also performed to reveal the genetic architecture of 10 traits. A lot of work has been done in this manuscript and they are quite impressive! However, some of analysis were not rigorously conducted and many of the points were not sufficiently discussed. My comments are as follows.

1) What is error rate of SNP calling? Are there any available SNP data set to compare with the SNPs developed in this paper? Are these SNPs segregating properly in mapping population? What is the allele frequency spectrum for these SNPs? What is the proportion of heterozygous genotypes of SNPs? Are these SNPs following HWE expectation? Were these SNPs imputed? If only phasing was done, what is the error rate of phasing (HAPCUT can give some estimates)? What is error rate of imputation (by masking known genotypes)? I understand not all the questions can be answered, but the authors need to discuss the quality before looking into biological questions.

2) The authors identified multiple independent domestication events. Possibly, this is due to 1. Low quality of SNPs. 2. Hybridization with wild mei as the authors mentioned. 3. Independent domestication events.

3) Line 99-101, I don't understand that statement. It needs to be rephrased. Samples with same phenotype are not expected to have many different SNPs (or genotype).

4) Figure 1. The phylogenetic tree does not have tree length or confidence value (bootstrap). I have troubles believing it.

5) Line 102-103, LD and genetic diversity are two different concepts.

6) Line 106, what's the half-maximum? I would suppose it is 1. I don't think this is a good way to describe r^2 . You can say r^2 decays to 0.1 in X bp. Also, I don't think conclusion of limited bottleneck is justified just by comparing LD with other species. The π need to be compared in mei and its ancestors.

7) What's the quality of the assembly of pan-genome? Are there any BAC sequences to check the quality of assembly? Or can the authors assemble the reference genome using the same approach and see how much is consistent between the two?

8) How many PAVs come from the unassembled sequences from other genomes?

9) Line 159-167, the different depth of PAV in different samples may come from sampling bias. There may be more samples coming from east China.

10) How was the population structure was controlled for in GWAS? It was not mentioned in the logistic regression. I didn't see a QQ plot for GWAS. Many of Manhattan plots have highly significant P-Value ($< 1E-20$), given the small sample size (>300), I have a feeling that these may come from population structure.

Reviewer #2 (Remarks to the Author):

This is a population genomics plus GWAS study using a diverse set of Mei and resequencing. Authors appear to have the germplasm knowledge, and the sequencing and analysis followed the established protocols. Unlike many crop species where domestication studies are well known, this study add new interesting information about ornamental woody plant domestication and trait dissection. I have the following comments.

Domestication.

- 1) It was not clear that why authors think that they can make statements about domestication from the floral trait other traits measured. If a combination of these traits can be used as the key indicator domestication traits (domestication syndrome traits as referred in cereals), one would expect a good separation of wild accessions versus landraces. This does not seem to be the case. Is it possible to analyze the trait data alone to identify the trait value combinations that generally separate wild from landraces (+cultivar)?
- 2) Figure 1a is difficult to see the spread of wild accessions among landraces. A PCA plot should be done to color code the different types (landraces, wild, and hybrid apricot-mei cultivar).
- 3) Figure 1b need to use the exactly same trait names and the same order as presented in legend. Also check the text.
- 4) This statement, "propose that several domestication events occurred", is well justified given the limited analysis done at this time.

Statistical analysis

- 1) GWAS method needs to be justified. The current description is not clear. This appears to be a case of testing of genotype frequency association with different trait classes. So, Chi-square test or other related tests, and the logistic regression would work. I am almost sure this been done, so no need to reinvent the wheel.
- 2) Please note "interval-mapping" is for linkage mapping/analysis with designed populations (biparent or four-way cross as indicated in Reference 58, 59), which is different from association mapping. I suspect authors followed the statistical tests for categorical phenotypes, but not the "interval-mapping", which constructs the test statistic for each incremental interval of 1 or 2 cM. The statement, "compared with interval mapping, the method we developed ..." is wrong and assumes extra burden of proof. Authors need to do thorough a literature search.
- 3) Please provide the citation for distance calculation. Again, search the literature and follow the procedure.
- 4) Reconstructing a phylogenetic tree using the SNPs within the DEGs identified for the traits appears to be a loop. Contrasting trait values/classes from groups of individuals to identify DEGs, and then SNPs within DEGs to cluster the individuals. This needs to be removed, or justified (possible to do so?).
- 5) I imagine that only a few genes would be underlying the floral color traits and was expecting to see some more specific GWAS signals, particularly due to the categorical nature of the trait. Was this due the fact that only one-dimensional scan was done?

Writing

- 1) The word "phenotype" should be replaced with something else in many places of the manuscript. The confusion is most obvious in the Figure 1 legend, where both "phenotype" and "trait" were used. "10 representative phenotypes" is not the same as "several representative phenotypes of these 10 traits". Please carefully revise throughout the manuscript.
- 2) Please clearly indicate that RNA-Seq was done for "two" accessions in the section header. Also,

indicate the phenotypes of the two. What is MDL and WYY?

3) L97-100. This statement is not clear. This may be due to the wrong use of "phenotypes". Use "classes" or a different word.

4) L109-100. This statement is not clear. Same reason as the last question?

5) I think the title is not justified by the current results.

Data access

Release of the data is critical for this type of research to have meaningful impact.

Reviewer #3 (Remarks to the Author):

This is review for article ID : NCOMMS-16-27657-T and entitled 'The Genetic Architecture of Floral Traits in the Woody plant *Prunus mume*' submitted to Nature Communications Journal.

Briefly, the present paper is notably reporting the identification of numerous loci associated to floral traits (i.e. petal color) in a collection of nearly 350 accessions of *P. mume*. To achieve this objective, authors analysed the sequencing data obtained from this set of accessions and conducted a population genetic analysis (population structure and variant analysis) and genome-wide association. Then, associated chromosomal regions were screened to identify potential candidate genes while the addition RNAseq, on a limited set of accession, validated some of these genes. Over represented gene ontologies were identified supporting the results. Finally, from the analysis of the sequencing data, the authors are proposing an interesting investigation of the core/Pan genome of the *Prunus* genus.

The present article delivers a massive set of data for the community of woody plants geneticists and physiologists, while supporting the scenario of domestication that *Prunus* species experienced. Furthermore, it provides numerous candidate genes to be deeply investigated (from a functional point of view).

From a technical point of view, to my knowledge, the genome sequencing and RNA sequencing were properly conducted and analysed. The population genetic analysis seems to be properly analysed. I'm aware about the limitation of the MS text size, but major information is not provided in the present version of the MS.

From an analytical point of view, the MS lacks of clarity: some methods reported here are nebulous and not appropriate. Hereafter, I'm developing the limits of the approach conducted here:

(1) The authors conducted a population ancestry/structure study to infer the number of genetic groups within the set of accessions. From this analysis, 16 groups were identified. However, if I'm right, population structure (Q) is a confounding effect in GWA analysis and has to be taken into account to avoid false positive association. Same observation was made for kinship (relatedness) between individuals (K). However, the authors do not report the implementation of such effects (K+Q) in the GWA model used here. Thus I consider that the GWA approach used in the present MS (linear regression) is not appropriate. Otherwise, authors have to develop much more their approach.

(2) Another effect of control for in GWA studies is multiple testing. Despite conducting a linear regression approach, such a correction has to be applied to correct the pvalues (HDR or Bonferroni).

(3) Some of the phenotypes used here are unordered categorical data explaining why the authors attempted to take this type of data into account for the GWA. However, their explanation is nebulous. From line 415 to line 418, I do not understand this sentence and the underlying objective. Please be more careful and add clarity to the M&M section.

(4) Conducting a GWA approach using categorical data while taking into account confounding effects

(i.e. K, Q) has been applied in human and other species. I would suggest revising this analytical approach to do so. The pioneer paper of Malosetti et al. (2007, *Genetics* 175(2), 879-889) is good starting point.

(5) Others points need a great clarification such as the LD computation or the methods underlying the GO analysis. There is no report of the methods. Adding a supplementary material file seems essential. For example, we do not know which metric was used to estimate LD decay (r^2 , D' ?) and if a threshold ($r^2 < 0.2$) was applied. Similarly, the identification of over-represented gene ontologies in chromosomal regions associated to floral traits has to be documented. For example, what is the percentage of genes annotated in the *Prunus* genomes? This would support the identification of the candidate genes reported.

Globally, I was frustrated by the discussion, which is too short. In the introduction, the authors claim that the domestication process remains unclear in woody plant but do not develop this point in the light of the results they are reporting. Thus, I would suggest developing the discussion and notably the pro and cons of the approach while valuing the original results obtained here.

Thus, I consider that the present paper is not suitable for publication in its present form and has to be revised, notably to adapt the GWA approach.

To the Editor and Reviewers:

We would like to thank the editor and reviewers for careful reading, and constructive suggestions for our manuscript. According to comments from the reviewers, we have comprehensively revised our manuscript. We think we have addressed all the questions mentioned by the reviewers thus we are resubmitting our revised manuscript. First, we have conducted analysis to prove the quality of SNPs and pan-genome assembly. Secondly, we have modified methods of the population analysis, especially the GWAS method. We applied a fixed-effects regression model using population structure as fixed covariate, thus we have updated analysis results and Method part in the revised manuscript. We are looking forward to your feedbacks and further comments/suggestions are welcome.

Below, we included the point-to-point response to the comments of the reviewers.

Reviewer #1

Mei is an ornamental woody plant in East Asia. The authors selected and sequenced 348 accessions to conduct the comprehensive diversity study of mei. In addition, a total of nine accessions were used to construct the pan-genome via deep sequencing. GWAS was also performed to reveal the genetic architecture of 10 traits. A lot of work has been done in this manuscript and they are quite impressive! However, some of analysis were not rigorously conducted and many of the points were not sufficiently discussed. My comments are as follows.

Response:

Thanks for the positive comments.

1)

- What is error rate of SNP calling? Are there any available SNP data set to compare with the SNPs developed in this paper?

Response:

We agree with the reviewer that the accuracy of SNP calling is important. However,

to our knowledge, there were no available SNP dataset to directly estimate error rate. Thus, accuracy of these SNPs can be reflected in the following aspects: 1) we applied widely used method to call the SNPs, where software (BWA+GATK) was applied in many plant resequencing project¹⁻³; 2) we also used the sequencing data of the same individual with the reference genome constructed to estimate the error rate. Using the same SNP-calling method, we identified ~0.28% sites to be homozygous but had the different genotype comparing to the reference, which maybe result from errors of the SNP calling or assembly, indicating the error rate should be less than 0.28%.

- Are these SNPs segregating properly in mapping population? What is the allele frequency spectrum for these SNPs?

Response:

Yes, all these SNPs are the segregating SNPs with a proper allele frequency spectrum. Allele frequency spectrum shows that minor alleles range from 0.06 to 0.49, with the distribution shown as following, indicating SNPs we used are segregating properly.

Rebuttal Figure 1. Minor allele frequency spectrum of population resequencing of mei

- What is the proportion of heterozygous genotypes of SNPs? Are these SNPs following HWE expectation?

Response:

There are about 97% SNPs with heterozygous genotypes (with more than one heterozygous sample), and over 50% of the SNPs has a heterozygous samples rate smaller than 0.05. The proportion of heterozygous samples rate of SNP is shown as below. About 59.77% raw SNPs obey the HWE expectation (P-value<0.01). We used maf<0.05 and hwe<0.01 to filter SNP in the downstream analysis such as GWAS.

Rebuttal Figure 2. Statistics of heterozygous rate of population resequencing of mei

- Were these SNPs imputed? If only phasing was done, what is the error rate of phasing (HAPCUT can give some estimates)? What is error rate of imputation (by masking known genotypes)?

Response:

Since the average sequencing depth of mei individuals is about 19.3X which is adequate for variants detection, thus we didn't carry out imputation. And the 5.34 million high quality SNPs we used were filtered with missing rate 0.1, which means that more than 90% individuals have determined genotypes. We also did not carry out phasing for these data set because the downstream analysis did not use such information.

2) The authors identified multiple independent domestication events. Possibly, this is due to 1. Low quality of SNPs. 2. Hybridization with wild mei as the authors mentioned. 3. Independent domestication events.

Response:

Yes, we agree to this comment. Since the quality of SNPs should be high, we think multiple independent domestication events should be attributed to reason 2 and 3.

3) Line 99-101, I don't understand that statement. It needs to be rephrased. Samples with same phenotype are not expected to have many different SNPs (or genotype).

Response:

Following the reviewer's suggestion, we have rephrased this sentence to: We investigated 10 representative ornamental traits of mei based on phylogenetic tree and found that samples with the same class often failed to cluster in one branch, indicating that it would be difficult to genetically distinguish samples with diverse traits using all genome-wide SNPs.

4) Figure 1. The phylogenetic tree does not have tree length or confidence value (bootstrap). I have troubles believing it.

Response:

According to this constructive suggestion, we have calculated bootstrap values for each node with 91.1% nodes (318/349) having a bootstrap value over 90 and the bootstrap values of main clades are high-confident.

5) Line 102-103, LD and genetic diversity are two different concepts.

Response:

Sorry for the misleading description, we have revised the sentence to mention LD instead of the genetic diversity.

6) Line 106, what's the half-maximum? I would suppose it is 1. I don't think this is a good way to describe r^2 . You can say r^2 decays to 0.1 in X bp. Also, I don't think

conclusion of limited bottleneck is justified just by comparing LD with other species. The Pi need to compared in mei and its ancestors.

Response:

We rephrased “half-maximum value” to “half of the highest value (highest value of different populations in our study varied from 0.458 to 0.577)” in the revised manuscript. We also calculated Pi for wild and domesticated mei in each 100kb window. Average Pi of wild mei is slightly lower than that of cultivated mei (0.0038 vs 0.0043), indicating that a domestication bottleneck in mei population may occur, or distant hybridization with other *Prunus* species may contributes to cultivar’s genetic diversity.

7) What’s the quality of the assembly of pan-genome? Are there any BAC sequences to check the quality of assembly? Or can the authors assemble the reference genome using the same approach and see how much is consistent between the two? (self-alignment)

Response:

Since no BAC sequences were available currently, we used other method for quality check of pan-genome assembly. We mapped the sequencing reads against the assembled genomes of each species. The distributions of sequence depth and GC depth are illustrated as follows, suggesting that the sequencing and the assembly meet the requirement of a further analysis. We assemble the reference genome using the same approach and the consistent ratio between the two is about 98.13%.

Rebuttal Table 1. Sequence depth distribution and the relation of GC frequency and depth average of each genome assembly

Sample ID	Sequence Depth	GC Depth
S93		

8) How many PAVs come from the unassembled sequences from other genomes?

Response:

As for the reviewer’s concern, we mapped sequencing reads from each individual and calculated sequence coverage to estimate proportion of PAV missing from assembly but covered by sequencing reads. About 6.25% (1,475 out of 23,590) of PAV resulted from unassembled sequences. We also filter unassembled sequences (often with length less than 100bp) in our final PAV set. We rephrased the method for identifying PAV in the revised manuscript. Each assembly were aligned to any other assemblies to identify any sequences specific to individuals (identity < 90%, length > 100 bp). We further eliminated PAVs with too many N’s (>10%) and with length less than 100bp.

9) Line 159-167, the different depth of PAV in different samples may come from sampling bias. There may be more samples coming from east China.

Response:

We agree with the reviewer that sampling bias might affect the identification of PAVs, in which more samples should result in more PAVs. However, here we are mentioning about examples of PAVs specific for some sub-populations. Furthermore, there are fewer samples from east China (**Supplementary Table 11b**).

10) How was the population structure controlled for in GWAS? It was not mentioned in the logistic regression. I didn't see a QQ plot for GWAS. Many of Manhattan plots have highly significant P-Value ($< 1E-20$), given the small sample size (>300), I have a feeling that these may come from population structure.

Response:

We agree with reviewer that the previous method had flaws for population structure. Thus, we have justified GWAS method and reanalyzed the data. To eliminate effects of population structure in GWAS, we have used population structure as fixed covariate in the regression model. More details of GWAS method have been added in the revised manuscript. QQ plots for P-values of each trait have been drawn, including no correction (green points) and correction for population structure (blue points). As you mentioned, highly significant P-values ($< 1E-20$) for many of Manhattan plots result from the confounding effect of population structure. We also found that no correction population structure can increase the false positive of GWAS for certain phenotypes from QQ plot. Thus, all results reported here are those after adjust for population structure. We also found trait-associated genes mentioned in the manuscript and using adjusted approach are mostly consistent.

a

b

c

d

e

f

Rebuttal Figure 3. QQ plots for GWAS p-values of 10 traits.

Reviewer #2

This is a population genomics plus GWAS study using a diverse set of Mei and resequencing. Authors appear to have the germplasm knowledge, and the sequencing and analysis followed the established protocols. Unlike many crop species where domestication studies are well known, this study add new interesting information about ornamental woody plant domestication and trait dissection. I have the following comments.

Response:

Thanks for the general positive comments.

Domestication.

1) It was not clear that why authors think that they can make statements about domestication from the floral trait other traits measured. If a combination of these traits can be used as the key indicator domestication traits (domestication syndrome traits as referred in cereals), one would expect a good separation of wild accessions versus landraces. This does not seem to be the case. Is it possible to analyze the trait data alone to identify the trait value combinations that generally separate wild from landraces (+cultivar)?

Response:

Thanks to the constructive suggestion, we used several combinations of traits in order to separate wild from landraces. We found that more traits didn't mean better separation (**Rebuttal Figure 4a**). However, by analyzing trait data of receptacle, stalk length, flower diameter and calyx top pattern, the spread of wild accessions among landraces is not so clear, but we can separate different cultivars and landraces (**Rebuttal Figure 4b, 4c**).

a

Rebuttal Figure 4. (a) PCA of all 30 traits data measured in this study. (b) (c) PCA of traits data of receptacle, stalk length, flower diameter and calyx top pattern, (c) different cultivars were color coded.

2) Figure 1a is difficult to see the spread of wild accessions among landraces. A PCA plot should be done to color code the different types (landraces, wild, and hybrid apricot-mei cultivar).

Response:

We performed PCA for all re-sequencing samples using genome-wide SNPs. PCA plot for first and second principle component was shown below. Wild mei are clustered (red circle) together but cannot be distinguished from cultivar.

Rebuttal Figure 5. PCA of SNPs generated from 348 mei resequencing data.

3) Figure 1b need to use the exactly same trait names and the same order as presented in legend. Also check the text.

Response:

Sorry for the mistake. We have modified these trait names and orders in Figure 1b in the revised manuscript. The current legend is as follows: The inner phylogenetic tree contains 16 subtrees and one outgroup. Different colors represent different subtrees. The clade color corresponds to the color of the outer circle with sample ID. The intermediate circles from the inner circle to the outer circle represent the traits petal color, stigma color, calyx color, bud color, staminal filament color, wood color, petal number, pistil character, bud aperture and branching phenotype. The color in each circle represents the phenotype of the trait. b) Several representative phenotypes of these 10 traits.

4) This statement, “propose that several domestication events occurred”, is well justified given the limited analysis done at this time.

Response:

We agree to this comment, but have just given a discussion about this issue. We rephrased the sentence to: our results suggest several domestication events may occur.

Statistical analysis

1) GWAS method needs to be justified. The current description is not clear. This appears to be a case of testing of genotype frequency association with different trait classes. So, Chi-square test or other related tests, and the logistic regression would work. I am almost sure this been done, so no need to reinvent the wheel.

Response:

We have justified GWAS method. A fixed-effects regression model using population structure as fixed covariate was tested relationships between genotypes and phenotypes. We used three different methods to detect association based logistic regression model for discrete traits. 1) General linear model (GLM), 2) Multinomial logistic model (MLM), 3) Ordinal logistic regression (OLR). For the continuous traits, linear regression model was used to test association.

We have also added more details for GWAS in this revised manuscript.

2) Please note “interval-mapping” is for linkage mapping/analysis with designed populations (biparent or four-way cross as indicated in Reference 58, 59), which is different from association mapping. I suspect authors followed the statistical tests for categorical phenotypes, but not the “interval-mapping”, which constructs the test statistic for each incremental interval of 1 or 2 cM. The statement, “compared with interval mapping, the method we developed ...” is wrong and assumes extra burden of proof. Authors need to do thorough a literature search.

Response:

This is a reasonable comment. This has been made clear. In this revision, we re-described the method of GWAS. Indeed, we only tested association between SNPs and phenotypes

in the whole genome, which belongs to association mapping rather than interval mapping. For categorical phenotypes, we conducted association mapping based logistical regression model.

3) Please provide the citation for distance calculation. Again, search the literature and follow the procedure.

Response:

We have added more details and citations for distance calculation in revised manuscript.

4) Reconstructing a phylogenetic tree using the SNPs within the DEGs identified for the traits appears to be a loop. Contrasting trait values/classes from groups of individuals to identify DEGs, and then SNPs within DEGs to cluster the individuals. This needs to be removed, or justified (possible to do so?).

Response:

As for the referee's concern, by constructing phylogenetic tree using the SNPs significantly related to specific traits, we intended to see whether these SNPs were able to distinguish samples from different traits, because phylogenetic tree using all SNPs showed a poor segregation of these traits. Besides, reconstructing of phylogenetic tree can also identify different sub-population, such as Fig 4b.

5) I imagine that only a few genes would be underlying the floral color traits and was expecting to see some more specific GWAS signals, particularly due to the categorical nature of the trait. Was this due the fact that only one-dimensional scan was done?

Response:

This is an important suggestion. Yes, we only perform one-dimensional scan due to the fact that over 5 million SNPs were involved in this study. Due to too many SNPs were involved, A two-dimensional scan will be resource-consuming. It will be significant to investigate in future study.

Writing

1) The word “phenotype” should be replaced with something else in many places of the manuscript. The confusion is most obvious in the Figure 1 legend, where both “phenotype” and “trait” were used. “10 representative phenotypes” is not the same as “several representative phenotypes of these 10 traits”. Please carefully revise throughout the manuscript.

Response:

We carefully rephrased “traits” and “phenotype” in the revised manuscript.

2) Please clearly indicate that RNA-Seq was done for “two” accessions in the section header. Also, indicate the phenotypes of the two. What is MDL and WYY?

Response:

As for the referee’s concern, we have deleted “RNA-Seq” from the section header to avoid misleading, based on the facts that RNA-Seq analysis was actually acting as a validation for GWAS results. Phenotypes of these two cultivars and the full descriptions of the abbreviations *Midanlv* (MDL) and *Wuyuyu* (WYY) have been supplemented in the revised manuscript.

3) L97-100. This statement is not clear. This may be due to the wrong use of “phenotypes”. Use “classes” or a different word.

Response:

Sorry for the misleading. We have rephrased this sentence in the revised manuscript to: We investigated 10 representative ornamental traits of mei based on phylogenetic tree and found that samples with the same class often failed to cluster in one branch, indicating that it would be difficult to genetically distinguish samples with diverse traits using all genome-wide SNPs.

4) L109-100. This statement is not clear. Same reason as the last question?

Response:

As for the referee’s concern, we have replaced “phenotype” with “trait” in the revised manuscript.

5) I think the title is not justified by the current results.

Response:

We have conducted whole genome resequencing of mei as a woody plant, and identified the whole genome diversity. Moreover, we conducted GWAS based on the genome wide SNPs, and identified associated SNPs/genes of the floral traits. Thus, we think we have depicted the genome architecture for mei and especially for the floral traits.

Data access

Release of the data is critical for this type of research to have meaningful impact.

Response:

We have uploaded the raw sequencing reads to NCBI under accession number SRA: SRP093801 (BioProject: PRJNA352648). We have included this information in the revised manuscript. The data is now under hold but will be publicly available after the publication.

Reviewer #3

Briefly, the present paper is notably reporting the identification of numerous loci associated to floral traits (i.e. petal color) in a collection of nearly 350 accessions of *P. mume*. To achieve this objective, authors analysed the sequencing data obtained from this set of accessions and conducted a population genetic analysis (population structure and variant analysis) and genome-wide association. Then, associated chromosomal regions were screened to identify potential candidate genes while the addition RNAseq, on a limited set of accession, validated some of these genes. Over represented gene ontologies were identified supporting the results. Finally, from the analysis of the sequencing data, the authors are proposing an interesting investigation of the core/Pan genome of the *Prunus* genus.

The present article delivers a massive set of data for the community of woody plants geneticists and physiologists, while supporting the scenario of domestication that *Prunus*

species experienced. Furthermore, it provides numerous candidate genes to be deeply investigated (from a functional point of view).

From a technical point of view, to my knowledge, the genome sequencing and RNA sequencing were properly conducted and analysed. The population genetic analysis seems to be properly analysed. I'm aware about the limitation of the MS text size, but major information is not provided in the present version of the MS.

Response:

Thanks for the general positive comments.

From an analytical point of view, the MS lacks of clarity: some methods reported here are nebulous and not appropriate. Hereafter, I'm developing the limits of the approach conducted here:

(1) The authors conducted a population ancestry/structure study to infer the number of genetic groups within the set of accessions. From this analysis, 16 groups were identified. However, if I'm right, population structure (Q) is a confounding effect in GWA analysis and has to be taken into account to avoid false positive association. Same observation was made for kinship (relatedness) between individuals (K). However, the authors do not report the implementation of such effects (K+Q) in the GWA model used here. Thus I consider that the GWA approach used in the present MS (linear regression) is not appropriate. Otherwise, authors have to develop much more their approach.

Response:

According to the comments of the reviewers, we have modified GWAS method. A fixed-effects regression model using population structure (Q) as fixed covariate was tested association. We have added more details for GWAS in this revised manuscript. We found correction population structure can obviously decrease the false positive of GWAS for the certain phenotypes from QQ plot. Due to property of nature population (no correlation between individuals), we do not use the kinship as random effects in regression model.

Rebuttal Figure 6. QQ plots for GWAS p-values of stigma color (a) and branching phenotype (b).

(2) Another effect of control for in GWA studies is multiple testing. Despite conducting a linear regression approach, such a correction has to be applied to correct the p-values (HDR or Bonferroni).

Response:

To avoid false positive association, we used $2.72e-9$ as the critical value which is calculated by Bonferroni correction at the 1% significance level in this GWAS study.

(3) Some of the phenotypes used here are unordered categorical data explaining why the authors attempted to take this type of data into account for the GWA. However, their explanation is nebulous. From line 415 to line 418, I do not understand this sentence and the underlying objective. Please be more careful and add clarity to the M&M section.

Response:

We have fixed and added more details for GWAS in this revised manuscript. Currently we conducted association mapping based logistical regression model.

(4) Conducting a GWA approach using categorical data while taking into account confounding effects (i.e. K, Q) has been applied in human and other species. I would suggest revising this analytical approach to do so. The pioneer paper of Malosetti et al. (2007, *Genetics*175(2), 879-889) is good starting point.

Response:

We tested association based logistical regression model for the binary, multinomial and order categorical data. To avoid confounding effects of population structure (Q) in GWAS, we using population structure as fixed covariate in the regression model. For the binary categorical data, the kinship between individuals can be used as random effects in mixed effect model based the binary logistical model, for example, the function glmer from the R package lem4 can complete the above work. For the multinomial and order categorical data, we did not find the appropriate GWAS method in mixed effect model based logistical model. Due to property of nature population, we do not use the kinship in this GWAS study.

(5) Others points need a great clarification such as the LD computation or the methods underlying the GO analysis. There is no report of the methods. Adding a supplementary material file seems essential. For example, we do not know which metric was used to estimate LD decay (r^2 , D' ?) and if a threshold ($r^2 < 0.2$) was applied. Similarly, the identification of over-represented gene ontologies in chromosomal regions associated to floral traits has to be documented. For example, what is the percentage of genes annotated in the Prunus genomes? This would support the identification of the candidate genes reported.

Response:

We rephrased the method for LD computation and GO analysis in the revised manuscript. The correlation coefficient (r^2) of alleles was calculated to measure LD decay in mei, and plotted without applying a threshold. We have added functional enrichment results in the Supplementary Table 26 for floral traits in the revised manuscript, and the percentage of genes annotated in mei genomes for each ontology were shown in column 3.

Globally, I was frustrated by the discussion, which is too short. In the introduction, the authors claim that the domestication process remains unclear in woody plant but do not develop this point in the light of the results they are reporting. Thus, I would suggest developing the discussion and notably the pro and cons of the approach while valuing the original results obtained here.

Response:

Thanks for the suggestion. We have carefully rephrased the discussion part in the revised manuscript.

Thus, I consider that the present paper is not suitable for publication in its present form and has to be revised, notably to adapt the GWA approach.

Response:

We carefully revised manuscript according to your suggestions, especially GWA approach where a fixed-effects regression model using population structure as fixed covariate was applying to test association. We're looking forward to your further comments on this version of manuscript.

References

- 1 Wang, M. *et al.* The genome sequence of African rice (*Oryza glaberrima*) and evidence for independent domestication. *Nat Genet* **46**, 982-988, doi:10.1038/ng.3044 (2014).
- 2 Zhou, Z. *et al.* Resequencing 302 wild and cultivated accessions identifies genes related to domestication and improvement in soybean. *Nat Biotechnol* **33**, 408-414, doi:10.1038/nbt.3096 (2015).
- 3 Wang, M. *et al.* Asymmetric subgenome selection and cis-regulatory divergence during cotton domestication. *Nat Genet* **49**, 579-587, doi:10.1038/ng.3807 (2017).

Reviewers' comments:

Reviewer #1 (Remarks to the Author):

Comments for the manuscript entitled "The genetic architecture of floral traits in the woody plant *Prunus mume*"

The revised manuscript is much improved compared with the initial submission. The authors took most of my remarks into account and responded them in the letter/manuscript. However, there still remain some problems in the manuscripts. Here are my comments.

1. I would suggest the authors to submit revised manuscripts with markup during review process in the future. It would be much easier to find where the changes are made. Also, specifying the changes (Page and line) in the review response will be helpful.
2. Response to the remarks need to be reflected in the manuscript. It is totally fine to reject the remarks as long as the reasons are justified. However, changes need to be made in the manuscript when the comments are helpful. This is how reviews contribute to improving the quality of the paper.
3. In the response to my #1 comment, the authors aligned the sequencing reads of the reference to the reference genome itself. They found 0.28% sites are homozygous and have different genotype, which was considered to be a false positive rate of SNP calling. Please report it in the manuscript.
4. In the response to my #4 comment, the authors did not respond in the manuscript. I think the bootstrap value is important Figure 1, because the authors proposed a multi-origin domestication analysis. Tree length is used to illustrate the genetic distance between samples and it is not the Figure 1. I understand the circular chart looks great but it is not quite scientifically insightful. I would suggest the authors to make a supplementary figure for the relationship of landraces, wild, the other species. It should be a rooted (other species as root) tree with bootstrap values.
5. In the response to my #7 comment, the authors did not make any changes in the manuscript.
6. In line 90, the authors mentioned a single origin domestication hypothesis. However, the important citation is missing.
7. I do not think there are sufficient evidence supporting the multi-origin domestication hypothesis. In addition, the hypothesis seems conflicting with existing population genetic theories. 1) The wild has lower diversity than cultivars as the authors mentioned the response to #6 comment. The cultivars generally has smaller genetic pool than the wild. Even though there are introgression from the wild, the cultivar should still have lower diversity than progenitors. 2) The wild has higher LD than the cultivars. In general, the cultivar has higher LD due to the founder effect during the domestication bottleneck.
8. The authors did not control for population structure well in GWAS. From the QQ plot, it is obvious that results are inflated. I would suggest the Q+K model for the GWAS. The author did not describe how the covariate of population structure was calculated and added in the model. My guess is that something is going on in the population structure control, even the authors claimed using the Q model.
9. Line 105. Again, I would not use the so called "half-maximum" to define LD, since the value of "half-maximum" keeps changing in different populations, there are no way to compare populations effectively.

Reviewer #2 (Remarks to the Author):

Authors made some attempts to improve the analysis and the manuscript. This is clear in GWAS analysis, SNP frequency plots, and data access.

However, stating "we have addressed all the questions mentioned by the reviewers" is probably not a good choice. After going through the response letter carefully, I was not fully convinced that this is the

case. In many places, although authors did some analyses suggested by the reviewers, they did not resolve the underlying questions raised by the reviewers. This is the case for my questions Domestication 1), 2), and 4) (Sorry for this question, I meant to say "is not well justified given the limited analysis done at this time"); Statistical analysis 1), 4), and 5); and Writing 5).

Some questions were consistently raised across reviewers and authors need to do a better job in revising the analysis and the way the results are interpreted.

For example, the Q-Q plots are nothing close to we expect to see: still too much deviation from the expected line for $-\log P$ values in the range of < 3 .

For questions that additional analyses suggested by reviewers in the last round of review could not resolve, it is authors' responsibility to either conduct other analyses or adjust statements to reveal the situation. Simply leaving it as was is not acceptable in general, and may only be done for a very limited number of questions.

In addition, please pay close attention to the writing. For example, you still have in L442 a partial sentence, "However, in GWASs, a number of traits, which are either discrete or continuous."

Reviewer #3 (Remarks to the Author):

This a second review for MS NCOMMS-16-27657A.

First of all, I would like to thank the authors for the effort they put to revise their MS according to the reviewers comments. Regarding the comments I raised during the first round of revision, the authors properly reanalysed their dataset, notably to improve the GWA model while adding more details on the Material and Methods used. Thus, the results reported are now stronger.

REgarding the rebuttal letter, I would have enjoyed that the authors would precisely cite the line numbers that the authors have modified according to the reviewers comments or provide a version of their MS that contains the track changes, to avoid comparing both initial and revised version of the MS.

Overall, I do not have any major nor minor concerns about the present version of the MS. However, I still frustrated about the weak discussion.

To the Reviewers:

According to comments from the reviewers, we have made major updates and changes in the MS and carefully considered and answered all the suggestions/concerns raised by the reviewers. We believe the MS is much improved this time, thus we are resubmitting our revised manuscript. First, we performed introgression analysis which provided more insights into population evolution. Secondly, by re-analyzing population structure, we have updated our GWAS results which were quite satisfactory as shown in Q-Q plots. We look forward to your re-consideration of MS and further comments/suggestions are welcome.

Below, we included the point-to-point response to the comments of the reviewers.

Reviewer #1:

Comments for the manuscript entitled “The genetic architecture of floral traits in the woody plant *Prunus mume*”

The revised manuscript is much improved compared with the initial submission. The authors took most of my remarks into account and responded them in the letter/manuscript. However, there still remain some problems in the manuscripts. Here are my comments.

Response:

Thanks for the positive comments. Below we give a point-to-point response to the remaining problems and we believe issues raised by reviewer will be better addressed this time.

1. I would suggest the authors to submit revised manuscripts with markup during review process in the future. It would be much easier to find where the changes are made. Also, specifying the changes (Page and line) in the review response will be helpful.

Response:

Thanks to the constructive suggestion. In this version, we carefully revised manuscript and kept remarks of all changes made this time and before.

2. Response to the remarks need to be reflected in the manuscript. It is totally fine to reject the remarks as long as the reasons are justified. However, changes need to be made in the manuscript when the comments are helpful. This is how reviews contribute to improving the quality of the paper.

Response:

We appreciate all comments and suggestions from the reviewer. In this version, we carefully revised manuscript which all responses were reflected in the main text or supplementary files.

3. In the response to my #1 comment, the authors aligned the sequencing reads of the reference to the reference genome itself. They found 0.28% sites are homozygous and have different genotype, which was considered to be a false positive rate of SNP calling. Please report it in the manuscript.

Response:

Following the reviewer's suggestion, we have added this sentence to the manuscript on line 80, page 3, "Applying the same variation calling method to the sequencing data of the same , individual of which the genome had been assembled as reference10, we identified 0.28% sites to be homozygous as a different genotype which might be false positive, indicating high accuracy of the variation calling method. "

4. In the response to my #4 comment, the authors did not respond in the manuscript. I think the bootstrap value is important Figure 1, because the authors proposed a multi-origin domestication analysis. Tree length is used to illustrate the genetic distance between samples and it is not the Figure 1. I understand the circular chart looks great but it is not quite scientifically insightful. I would suggest the authors to make a supplementary figure for the relationship of landraces, wild, the other species. It should be a rooted (other species as root) tree with bootstrap values.

Response:

We re-perform phylogenetic tree using genome-wide SNPs and added the figure to supplementary files (Supplementary Figure 2). A rooted tree with bootstrapping values and branch length were drawn at this time.

Rebuttal Figure 1. Phylogenetic tree of 351 resequencing individuals with outgroup as root and bootstrapping value for each branch.

5. In the response to my #7 comment, the authors did not make any changes in the manuscript.

Response:

Following the reviewer’s suggestion, we have added this sentence to the manuscript on page 6 line 190, “We found about 6.25% of PAVs resulted from unassembled sequences and thus excluded them from following analysis”.

6. In line 90, the authors mentioned a single origin domestication hypothesis. However, the important citation is missing.

Response:

We have added more citations in the manuscript, which mentioned mei domestication history and classification system. Different from many crop populations, most landraces in our population are selected from wild groups, thus they could be regarded as founders as well. Besides, we have made major updates in the genetic structure part. Interspecific admixture was identified in

cultivars which leads to the complicated genetic structure of mei. Our main findings here was that introgressions, especially interspecific introgressions were identified in some cultivar groups, which previously reported to have pure mei genetic basis.

7. I do not think there are sufficient evidence supporting the multi-origin domestication hypothesis. In addition, the hypothesis seems conflicting with existing population genetic theories. 1) The wild has lower diversity than cultivars as the authors mentioned the response to #6 comment. The cultivars generally have smaller genetic pool than the wild. Even though there are introgression from the wild, the cultivar should still have lower diversity than progenitors. 2) The wild has higher LD than the cultivars. In general, the cultivar has higher LD due to the founder effect during the domestication bottleneck.

Response:

We further excluded hybrids from our cultivar population, and only include SNP with maf > 0.05 and missing rate lower than 0.2 instead of using raw SNP data set, which we believe may lead to calculation bias. After filtration, average pi for wild and cultivar were 0.00282 and 0.00201, respectively. We mentioned the result in population evolution part.

We appreciate reviewer's suggestion about interpretation of LD results. According to our results, wild population has higher LD may due to weak domestication bottleneck, which is supported by similar genetic diversity between wild and cultivated population as well. Besides, as mentioned before, domestication history suggested many landraces in our population can be regarded as founders. We added more citations and descriptions in the population evolution part.

8. The authors did not control for population structure well in GWAS. From the QQ plot, it is obvious that results are inflated. I would suggest the Q+K model for the GWAS. The author did not describe how the covariate of population structure was calculated and added in the model. My guess is that something is going on in the population structure control, even the authors claimed using the Q model.

Response:

Thank you for your comment, but we do have controlled population structure. It is a good suggestion to use the Q+K model because it has proven powerful for controlling population structure in crops and other organisms. Our study population is composed of different landraces of a woody plant, whose selection has a different process from what is true in crops. Our landraces were mostly selected from wild populations, thus each sample in our population is regarded as a founder. As such, kinships (K) among these landraces are unknown.

For this reason, we used the Q model (fastStructure v1.0), published in 2014, to analyze population structure. First, we determined the optimal number of subpopulations among these landraces based on all filtered SNPs (that are subsequently used in GWAS) and assigned each landrace into its optimal subpopulation. Second, using these subpopulations as covariates, we performed GWAS analysis by a mixed-effect regression model. It turns out that these two steps provide quite satisfactory results about GWAS, as shown by Q-Q plots for p values for each trait (Rebuttal Fig. 5), which p-values after correction are found to be much close to expected curves.

9. Line 105. Again, I would not use the so called “half-maximum” to define LD, since the value of “half-maximum” keeps changing in different populations, there are no way to compare populations effectively.

Response:

We agree with the reviewer’s suggestion and delete these comparisons and descriptions in this version of revised manuscript.

Reviewer #2:

Authors made some attempts to improve the analysis and the manuscript. This is clear in GWAS analysis, SNP frequency plots, and data access.

However, stating “we have addressed all the questions mentioned by the reviewers” is probably not a good choice. After going through the response letter carefully, I was not fully convinced that this is the case. In many places, although authors did some analyses suggested by the reviewers, they did not resolve the underlying questions raised by the reviewers. This is the case for my questions Domestication 1), 2), and 4) (Sorry for this question, I meant to say “is not well justified given the limited analysis done at this time); Statistical analysis 1), 4), and 5); and Writing 5).

Response:

Thanks for reviewer’s suggestions. We have made major changes in population genetics and GWAS, by including updated introgression analysis, for detection of inter- and intro-specific admixture, and GWAS. We believe this time major issues should be addressed properly.

Domestication.

1) It was not clear that why authors think that they can make statements about domestication from the floral trait other traits measured. If a combination of these traits can be used as the key indicator domestication traits (domestication syndrome traits as referred in cereals), one would expect a good separation of wild accessions versus landraces. This does not seem to be the case. Is it possible to analyze the trait data alone to identify the trait value combinations that generally separate wild from landraces (+cultivar)?

Response:

Thanks to the constructive suggestion. This time we made major changes in population evolution part. We mentioned classification system of mei reported previously, which mainly considered morphological factor to classify domesticated mei into different groups. We did find that using traits alone couldn't get better separation between wild and cultivated mei (**Rebuttal Figure 2**). Phylogenetic analysis also indicated that landraces with similar phenotypes (or same cultivar group referring to classification system) often failed to clustered together in the phylogenetic tree. Taken together, we rephrased results of population evolution to avoid possible misleading descriptions.

Rebuttal Figure 2. PCA of all 30 traits data measured in this study

2) Figure 1a is difficult to see the spread of wild accessions among landraces. A PCA plot should be done to color code the different types (landraces, wild, and hybrid apricot-mei cultivar).

Response:

We reanalyzed genome-wide SNP and got a PCA plot using EIGENSOFT, a more commonly used software. We could see a clearer spread of wild and cultivated mei, meanwhile, we could see a separation between hybrid and landraces. We also color coded different population according to population structure result and most subpopulations could be clustered together in the PCA plot.

Rebuttal Figure 3. PCA of 351 resequencing samples.

4) This statement, “propose that several domestication events occurred”, is not well justified given the limited analysis done at this time.

Response:

We agree to this comment and have made major updates in the genetic structure part. To explain the genetic structure of mei, we made major changes in our manuscript in this part. Our study population is composed of different landraces of a woody plant, whose selection has a different process from what is true in crops. Our landraces were mostly selected from wild populations, thus each sample in our population is regarded as a founder. Besides, interspecific admixture was identified in cultivars. We perform introgression analysis for different cultivar populations of mei, and results have shown modern mei cultivar almost have different origin, including extensive introgression from other *Prunus* species to mei also shown from genomic view for the first time. We rephrased this part by adding more evidence and citations.

Statistical analysis

1) GWAS method needs to be justified. The current description is not clear. This appears to be a case of testing of genotype frequency association with different trait classes. So, Chi-square test or other related tests, and the logistic regression would work. I am almost sure this been done, so no need to reinvent the wheel.

Response:

Thank you for your comment. Different from crop populations, our landraces were mostly selected from wild populations, thus each sample in our population is regarded as a founder. As such, kinships (K) among these landraces are unknown.

For this reason, we used the Q model (fastStructure v1.0), published in 2014, to analyze population structure. First, we determined the optimal number of subpopulations among these landraces based on all filtered SNPs (that are subsequently used in GWAS) and assigned each landrace into its optimal subpopulation. Second, using these subpopulations as covariates, we performed GWAS analysis by a mixed-effect regression model. It turns out that these two steps provide quite satisfactory results about GWAS, as shown by Q-Q plots for p-values for each trait (Rebuttal Fig. 5), which p-values after correction are found to be much close to expected curves.

We have also added more details for GWAS in this version of revised manuscript.

4) Reconstructing a phylogenetic tree using the SNPs within the DEGs identified for the traits appears to be a loop. Contrasting trait values/classes from groups of individuals to identify DEGs, and then SNPs within DEGs to cluster the individuals. This needs to be removed, or justified (possible to do so?).

Response:

We agree with reviewer's suggestion and deleted phylogenetic tree using SNPs identified from GWAS and DEGs.

5) I imagine that only a few genes would be underlying the floral color traits and was expecting to see some more specific GWAS signals, particularly due to the categorical nature of the trait. Was this due the fact that only one-dimensional scan was done?

Response:

Thanks for reviewer's suggestion. Yes, we only perform one-dimensional scan considering our population (natural population) and due to the fact that over 5 million SNPs were involved in this study. Besides, after re-analyze and better control of the population structure, we got better results in GWAS which provided more reliable evidence and facilitated identification of major genes underlying those floral traits. In addition, combined with gene regulation network database (String), we were able to reveal more details towards the whole picture underlying those traits as well.

Writing

5) I think the title is not justified by the current results.

Response:

We made major changes in genetic structure as well as GWAS results. This time, we think we have more reliable results and more insights into genetic architecture for mei and especially for the floral traits.

Some questions were consistently raised across reviewers and authors need to do a better job in revising the analysis and the way the results are interpreted.

Response:

Thanks, we carefully revised each question and add more analysis to interpret the result.

For example, the Q-Q plots are nothing close to we expect to see: still too much deviation from the expected line for $-\log P$ values in the range of < 3 .

Response:

We agree with reviewer's suggestion. In this revision, we re-perform population structure for mei at first, using FastStructure (v1.0) published on 2014. Results are shown in Rebuttal Fig 4. and also updated in the Supplementary Figures. We then perform GWAS for each trait considering our newly updated population structure and QQ plots for p-values of each trait are drawn (Rebuttal Fig 5). From the plots, much better results could be seen, where p-values after correction are much closer to the expected curve.

Rebuttal Figure 4. Population structure of wild and cultivated mei.

a

b

c

d

e

f

Rebuttal Figure 5. QQ plots for GWAS p-values of 10 traits (a-j: petal number, petal color, stigma color, bud color, wood color, staminal filament color, pistil character, bud aperture, branching phenotype, calyx color).

For questions that additional analyses suggested by reviewers in the last round of review could not resolve, it is authors' responsibility to either conduct other analyses or adjust statements to reveal the situation. Simply leaving it as was is not acceptable in general, and may only be done for a very limited number of questions.

Response:

Thanks for all reviewers' suggestions aiming at making our research more available for whole scientific society. We believe this version will be much better than before, because we have made major changes in our manuscript and have carefully revised the manuscript according to the reviewers' suggestions.

In addition, please pay close attention to the writing. For example, you still have in L442 a partial sentence, “However, in GWASs, a number of traits, which are either discrete or continuous.”

Response:

Sorry for the mistake. We have carefully revised writing errors and major changes have been made for this time. The sentence you mentioned is now revised to “However, in GWASs, a number of traits are either discrete or continuous” on line 535, page 16.

Reviewer #3:

First of all, I would like to thank the authors for the effort they put to revise their MS according to the reviewers’ comments. Regarding the comments I raised during the first round of revision, the authors properly reanalysed their dataset, notably to improve the GWA model while adding more details on the Material and Methods used. Thus, the results reported are now stronger.

Response:

Thanks for reviewer’s positive comments.

Regarding the rebuttal letter, I would have enjoyed that the authors would precisely cite the line numbers that the authors have modified according to the reviewers comments or provide a version of their MS that contains the track changes, to avoid comparing both initial and revised version of the MS.

Response:

We agree with review’s suggestions, and keep tracks of all modifications from the original manuscript in this version.

Overall, I do not have any major nor minor concerns about the present version of the MS. However, I still frustrated about the weak discussion.

Response:

Thanks for reviewer’s suggestion. We strengthened discussion part in this version of manuscript. We believe this time it was much more enriched than before.

Reviewers' comments:

Reviewer #1 (Remarks to the Author):

Comments for the manuscript entitled "The genetic architecture of floral traits in the woody plant *Prunus mume*"

I would thank the authors for taking most of my comments into account and respond them properly in the manuscript. Globally, I would consider that the authors made considerable improvement compared to the previous version, however, there are still a couple of discussions did not convince me.

1. In the response to my #7 comment, the new result shows that the wild has higher genetic diversity and higher LD than the cultivars. The authors argue that this is due to weak domestication. I do not think that is the case. It is reasonable to have lower diversity in cultivars. However, higher LD in wild populations is not convincing, which is due to the fact random drift and selection after the bottleneck usually increase LD in cultivars. Is it possible that the passport data of the wild are not quite valid? Or the wild mei accessions (not apricot or plum hybrid) have introgression in them?

2. In the response to my #8 comment, the authors did not use Q+K model because the pedigree information is not available for mei. However, the marker based kinship can be estimated from SNP data. This is an obvious mistake.

To the Editor and Reviewer:

We are highly grateful for your time and effort given to our manuscript at different stages. We have carefully considered and answered both scientific and editorial suggestions/comments raised by the reviewer and editor. First, we looked deep into linkage disequilibrium of wild and cultivar populations, where we found higher LD in most cultivar populations. Secondly, we used Q as well Q+K models for each trait in GWAS, and optimal model was chosen according to the Q-Q plot, in conjunction with the inflation factor estimated from the median of the test statistics for all the markers. This time, we believe we have provided solid evidence for novel conclusions. We look forward to your reply and further comments/suggestions are welcome.

Below, we included the point-to-point response to the comments of the reviewer.

Reviewer #1

I would thank the authors for taking most of my comments into account and respond them properly in the manuscript. Globally, I would consider that the authors made considerable improvement compared to the previous version, however, there are still a couple of discussions did not convince me.

1. In the response to my #7 comment, the new result shows that the wild has higher genetic diversity and higher LD than the cultivars. The authors argue that this is due to weak domestication. I do not think that is the case. It is reasonable to have lower diversity in cultivars. However, higher LD in wild populations is not convincing, which is due to the fact random drift and selection after the bottleneck usually increase LD in cultivars. Is it possible that the passport data of the wild are not quite valid? Or the wild mei accessions (not apricot or plum hybrid) have introgression in them?

Response:

Thanks to this comment/suggestion, we have further investigated possible reasons for longer LD in the wild population comparing to the cultivated population.

First of all, we agree with the reviewer that incomplete sampling and possible introgression of cultivars to wild individuals might result in higher LD in wild population. For sampling of the wild population, since no previous complete survey of wild population was available, we only sampled 15 wild individuals according to the geographical information, in order to make sure major distribution areas of wild mei were covered. For the second possibility of introgression, we analyzed whether there were notable introgressions for all the 15 wild individuals (**Rebuttal Fig. 1**). We found no significant introgression from cultivars for at least 14 wild individuals, with only one individual (Sample ID: 329) with notable introgression from other wild population and other *Prunus* species. Thus, we think introgression might not be major reason for the higher LD in wild population, or to say, lower LD in cultivated.

Furthermore, we analyzed LD of sub-populations of cultivars (**Rebuttal Fig. 2, Supplementary Fig. 3c**) in order to find possible reason for low LD in the cultivated population. We found 5 out of the 7 sub-populations showed notably high LD comparing to the wild population, while *Gongfen* and *Danban* had substantially lower LD. Considering massive introgression from other species (**Fig. 2**), thus, the introgression of other species to these two cultivated sub-populations resulted in low LD, and also in the cultivated population with all cultivars. With the further analysis conducted here, we revised the manuscript accordingly.

Rebuttal Fig1. Introgression for wild population

Rebuttal Fig 2. Linkage disequilibrium between wild and all cultivar classes

2. In the response to my #8 comment, the authors did not use Q+K model because the pedigree information is not available for mei. However, the marker based kinship can be estimated from SNP data. This is an obvious mistake.

Response:

We agree with the reviewer that the degree of kinship between any two cultivars (which are virtually the founders in our case) can be estimated through correlation analysis using marker data. In fact, the genetic similarity of cultivars determined on the basis of marker data has already been used in the way all cultivars were clustered into distinct groups. If the study material contains population structure according to the information criterion, we will find more than one group among the material. All cultivars within each group tend to be similar in genetic relatedness. The Q model was used to remove the difference due to population structure. If the kinship, i.e., the most probable identity by state of each allele between cultivars within a group, affects the result about a trait's GWAS, then we prefer to use the Q+K model that can adjust for difference due to the kinship.

For this reason, we used both Q and Q+K models for each trait and select one that better explain the data based on Q-Q plot and the estimation of inflation factor. The results are summarized as follows:

Rebuttal Table 1. Statistics of optimal model for each trait

Trait	Optimal model
Petal number	Q+K
Petal color	Q+K
Stigma color	Q+K
Bud color	Q
Wood color	Q
Staminal filament color	Q
Pistil character	Q
Bud aperture	Q+K
Branch phenotype	Q
Calyx color	Q

From the Q-Q plot of staminal filament color, it is a dilemma to choose a more appropriate model. We further calculated the inflation factor (λ) that determines how much the estimation of the test statistic by a model is inflated. The estimated λ is 1.51 for the Q model and 1.73 for the Q+K model. Therefore, we choose the Q model because λ is closer to one.

In this version, we have presented GWAS results for petal number, petal color, stigma color and bud aperture based on the Q+K model. It is also seen that the SNPs detected by the Q+K model largely overlap those by the Q model. We accordingly updated the supplementary figure of Q-Q plot by using the result from optimal model.

Q Model

Q+K Model

Petal number

Petal color

Stigma color

Bud color

Wood color

Staminal filament color

Pistil character

Bud aperture

Branch phenotype

Calyx color

Rebuttal Fig 3. Q-Q plot of both Q and Q+K model for each trait

REVIEWERS' COMMENTS:

Reviewer #1 (Remarks to the Author):

Comments for the manuscript entitled "The genetic architecture of floral traits in the woody plant *Prunus mume*"

I am glad to see the authors made significant improvement on this version. My concerns were taken into account and the manuscript was properly revised. I just have one minor suggestion.

Extensive introgression was found in *mei*, which is probably the reason why higher LD was observed in cultivars. It would be great to put this in the discussion.

To the Editor and Reviewer:

We are highly grateful for all your effort given to our manuscript. We have carefully considered all comments/suggestions raised by the reviewer and editor. Below, we included the point-to-point response to the comments of the reviewer.

Reviewer #1 (Remarks to the Author):

Comments for the manuscript entitled "The genetic architecture of floral traits in the woody plant *Prunus mume*"

I am glad to see the authors made significant improvement on this version. My concerns were taken into account and the manuscript was properly revised. I just have one minor suggestion.

Extensive introgression was found in mei, which is probably the reason why higher LD was observed in cultivars. It would be great to put this in the discussion.

Response

Thanks a lot for reviewer's suggestion. We added following sentence in the Discussion section, "It suggested that introgression was probably the reason why higher LD was observed in wild population, as compared to that in these cultivar groups".